# SimpleVLA-RL: Scaling VLA Training via Reinforcement Learning

**Haozhan Li**[*1,3], **Yuxin Zuo**[*1], **Jiale Yu**[*1], **Yuhao Zhang**[*4], **Zhaohui Yang**[4], **Kaiyan Zhang**[1,7]
**Xuekai Zhu**[1], **Yuchen Zhang**[5], **Tianxing Chen**[6], **Ganqu Cui**[2], **Dehui Wang**[4]
**Dingxiang Luo**[4], **Yuchen Fan**[4], **Youbang Sun**[1], **Jia Zeng**[2], **Jiangmiao Pang**[2]
**Shanghang Zhang**[5], **Yu Wang**[1], **Yao Mu**[2,4], **Bowen Zhou**[†1,2], **Ning Ding**[†1,2]

[1]Tsinghua University   [2]Shanghai AI Lab   [3]Nature Will   [4]Shanghai Jiao Tong University
[5]Peking University   [6]The University of Hong Kong   [7]Frontis.AI

 **Code:** `https://github.com/PRIME-RL/SimpleVLA-RL`

## Abstract

Vision-Language-Action (VLA) models have emerged as a powerful paradigm for robotic manipulation. Despite substantial progress enabled by large-scale pretraining and supervised fine-tuning (SFT), these models face two fundamental challenges: (i) the scarcity and high cost of large-scale robotic trajectories required for SFT scaling, and (ii) limited generalization to tasks under distribution shift. To overcome these limitations, we explore reinforcement learning (RL) as a pathway to scaling VLA training beyond limited datasets. Inspired by LLM breakthroughs where RL with outcome rewards enhances step-by-step reasoning, we ask: ***Can outcome-driven RL improve long-horizon step-by-step action planning of VLA?*** In this work, we introduce **SimpleVLA-RL**, an efficient RL framework tailored for VLA models. Building upon veRL, we introduce VLA-specific trajectory sampling, scalable parallelization, multi-environment rendering, and optimized loss computation. Applied to OpenVLA-OFT, **SimpleVLA-RL** achieves 99% of SoTA performance on LIBERO and 80% relative improvement on RoboTwin 1.0&2.0, outperforming $\pi_0$ with our proposed exploration-enhancing strategies. **SimpleVLA-RL** reduces dependence on large-scale data, enables robust generalization, and remarkably surpasses SFT in real-world tasks. Moreover, we identify a novel phenomenon "pushcut" during RL training, wherein the policy discovers unseen patterns beyond those seen in previous training process.

## 1 Introduction

Vision-Language-Action (VLA) models have emerged as a promising approach for general robotic manipulation by integrating visual perception, language understanding, and action generation in a unified framework (Firoozi et al., 2025; Kim et al., 2024; Zhong et al., 2025). Current VLA training paradigm consists of two stages: large-scale pretraining on multimodal data (human manipulation videos (Sapkota et al., 2025), image-text pairs, and heterogeneous robot datasets (O'Neill et al., 2024)), followed by supervised fine-tuning (SFT) on additional high-quality robot trajectories to enhance task-specific capabilities.

While imitation learning paradigm has achieved notable progress (Intelligence et al., 2025), its heavy dependence on large-scale, high-quality data poses a fundamental bottleneck that constrains further development of VLA models:**the Generalization Bottleneck from Data Scarcity** (Schulman et al., 2017b). **1) Data Scarcity:** Scaling VLA training through SFT requires massive amounts of high-quality robot trajectories, yet such data remains scarce and prohibitively expensive (Gao et al., 2024). Collecting expert demonstrations demands carefully designed scenarios, diverse manipulation objects, and skilled operators, which severely constrains both data scale and diversity (Bu et al., 2025a; Team et al., 2025a). **2) Poor Generalization:** This data scarcity leads to a fundamental mismatch between training distributions and open-ended real-world tasks (Liu et al., 2025a). VLA

---

[*]Equal Contributions   [†]Corresponding Authors

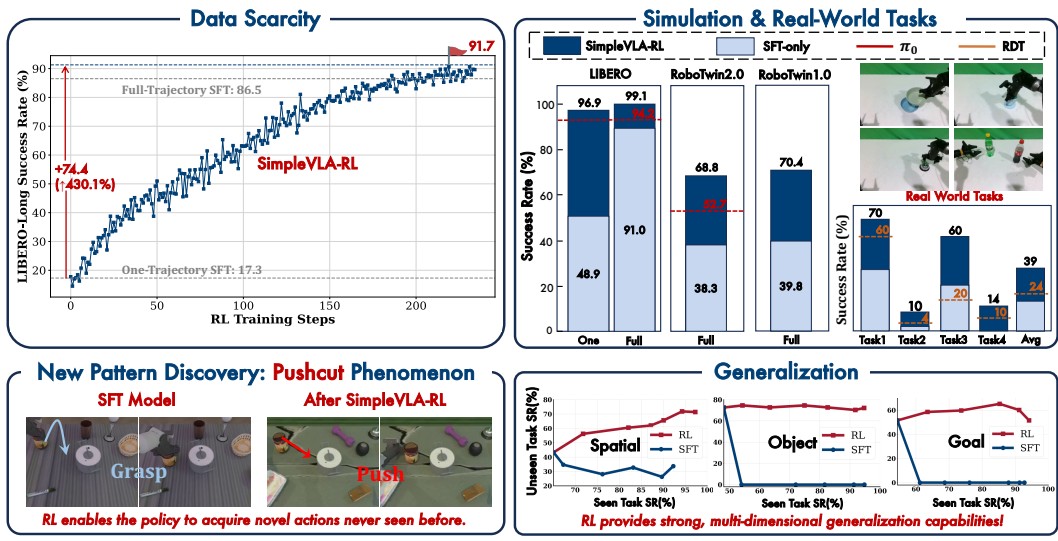

Figure 1: Overview of **SimpleVLA-RL**. An efficient RL framework for VLA that improves long-horizon planning under data scarcity, outperforms SFT in simulation and real-world tasks, reveals a "pushcut" new-action phenomenon, and strengthens spatial/object/goal generalization.

models trained on limited, scene-specific data tend to memorize patterns rather than learn generalizable skills. Consequently, even minor distribution shifts, unseen objects or environments, can cause compounding errors that severely limit generalization (Ross & Bagnell, 2010). This problem becomes especially critical in compositional and long-horizon tasks (Gupta et al., 2019), revealing that simply scaling SFT data cannot build generalizable VLA models to the open world.

To overcome this generalization bottleneck, VLA models need a learning mechanism capable of interactive refinement beyond limited static datasets. Reinforcement Learning (RL) offers this capability through trial-and-error environmental interaction (Xu et al., 2024). However, traditional robotics RL requires hand-crafted reward functions for each task, limiting scalability and generalization to novel scenarios where rewards are undefined (Ibarz et al., 2021; Kroemer et al., 2021; Ma et al., 2023). This creates a dilemma: SFT is constrained by data, while traditional RL is constrained by reward engineering. Recent breakthroughs in Large Reasoning Models (LRMs) provide a crucial insight: using only sparse outcome rewards, RL can significantly enhance models' ability to generate correct step-by-step reasoning chains (Guo et al., 2025a; Yang et al., 2025; Zeng et al., 2025; Team et al., 2025b). This raises a critical question for VLAs: **Can we leverage this outcome-driven RL paradigm to enhance VLA models' ability to generate step-by-step correct actions?** However, applying RL to VLA models presents unique technical challenges. First, unlike open-loop text generation in LLMs, VLAs require closed-loop environment interaction with continuous visual feedback, creating substantial computational overhead (Wang et al., 2025). Second, manipulation tasks face inefficient exploration due to high-dimensional action spaces and sparse rewards (Zhou et al., 2025). Third, existing RL frameworks lack VLA-specific infrastructure for efficient inference and parallel environment interaction (Luo et al., 2025).

We introduce **SimpleVLA-RL**, an effective RL framework for VLA models. Building upon Volcano Engine Reinforcement Learning for LLMs (veRL) (Sheng et al., 2024), we enable end-to-end rule-based online RL for VLA models through VLA-specific interactive trajectory sampling and loss computation. To support scalable RL training, we extend veRL with parallel multi-environment rendering and adapt it into a unified training–inference–rendering framework. We also design three exploration-enhancing strategies. Through **SimpleVLA-RL**, we significantly improve VLA performance under data-scarce conditions, enhance generalization capabilities, and achieve noticeable gains in real-world applications. Surprisingly, during RL training, the policy discovers novel patterns beyond those in supervised data, a phenomenon we term "pushcut". Our contributions include:

- **Efficient online RL framework for VLA:** We develop an efficient end-to-end VLA online RL framework based on veRL that enables stable, sample-efficient training, optimized for rendering parallelization and distributed training & inference.

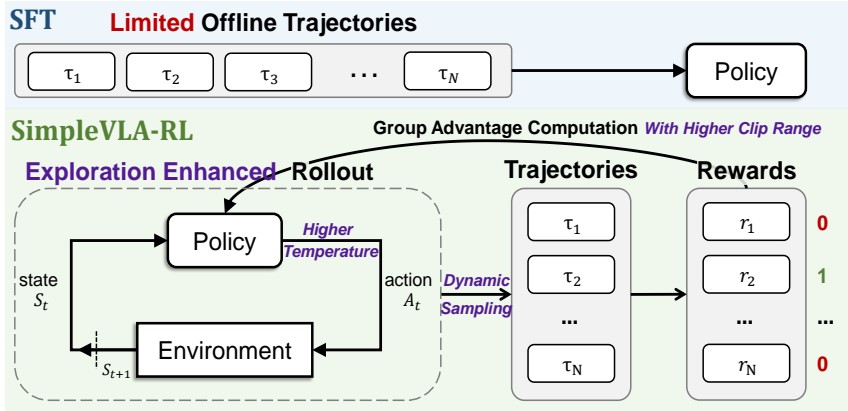

Figure 2: Overview of **SimpleVLA-RL**.

- **SOTA performance:** We incorporate exploration-enhancing strategies, yielding consistent performance improvements of 10–15%. Moreover, **SimpleVLA-RL** surpasses multiple SoTA baselines on both LIBERO and RoboTwin 1.0 & 2.0.

- **Data efficiency and generalization:** With only a single demonstration per task, RL boosts LIBERO-Long success rates from 17.1% to 91.7%, and significantly outperforms SFT in spatial, object, and task generalization.

- **Real-world deployment capability:** Simulation-trained policies transfer effectively to real-world, achieving strong sim-to-real improvements without requiring real robot data.

## 2 SIMPLEVLA-RL

DeepSeek-R1 (Guo et al., 2025a) has achieved remarkable performance gains through online RL with the simple, scalable rule-based reward, highlighting a promising training paradigm. In this section, we introduce **SimpleVLA-RL**, which extends this rule-based online RL framework to VLA models for embodied manipulation tasks as shown in Figure 2. Specifically, our training framework proceeds as follows: we begin by generating multiple trajectories for each input via random sampling (§**2.1**). Each trajectory is then assigned a simple outcome reward (1 for success, 0 for failure) based on environment feedback (§ **2.2**). Leveraging these rewards together with the corresponding action token probabilities, we compute the GRPO loss to update the policy model (§ **2.4**).

### 2.1 INTERACTIVE VLA ROLLOUT

RL on VLA models differs fundamentally from LLMs in trajectory generation. To enable online RL, policy models need to generate diverse trajectories from an input for effective exploration. While LLMs naturally achieve this diversity through random sampling on text token distributions (Renze, 2024; De Rosa & Papa, 2021), VLA models face a unique challenge due to their action decoding strategies. Current VLA models often employ three strategies: (1) generating action token distributions similar to LLMs (Black et al., 2024; Kim et al., 2024), (2) diffusion-based denoising on latent states (Liu et al., 2024; Cheang et al., 2025), and (3) deterministic regression via MLPs (Kim et al., 2025). Among these, the token-based approach is most compatible with PPO-like RL algorithms, as it naturally provides action distributions necessary for both random sampling and policy gradient computation. Therefore, we adopt this approach, where the VLA model outputs action token probability distributions and employs random sampling to generate diverse trajectories.

Furthermore, for a given input query, LLM rollout proceeds by autoregressively generating tokens until reaching a stop token or max output length. In contrast, VLA rollout requires continuous interaction with the environment to update the visual observation and robot state dynamically (as detailed in Appendix B). This closed-loop interaction is necessary because each robotic action alters the environment, and subsequent actions must be conditioned on real-time sensory feedback. We present the comparison of the rollout algorithm pseudo-code of LLMs and VLA in Listing 1.

## 2.2 OUTCOME REWARD MODELING

**SimpleVLA-RL** employs a straightforward binary reward function for RL training. Unlike traditional RL approaches that require carefully crafted reward functions (Hadfield-Menell et al., 2017; Knox et al., 2023; Booth et al., 2023), we follow DeepSeek-R1's approach by assigning trajectory-level rewards of either 0 or 1 based solely on task completion. When the VLA model successfully completes a task, the entire trajectory is assigned a reward of 1; otherwise, it receives a reward of 0. For gradient computation, these trajectory-level rewards are uniformly propagated to the individual action tokens. Consequently, all tokens within successful trajectories are assigned a reward of 1, whereas those in unsuccessful trajectories are assigned a reward of 0. Our reward function is:

$$R(a_{i,t} \mid s_{i,t}) = \begin{cases} 1, & \text{is\_successful}[\text{traj}_i(a_i, s_i)], \\ 0, & \text{otherwise.} \end{cases} \tag{1}$$

This simple outcome-level reward is simple yet effective: scalable, broadly applicable across environments, and free from complex process-based design (Wu et al., 2021). By focusing solely on task completion, it avoids the non-transferability issues typical of task-specific rewards.

## 2.3 EXPLORATION ENHANCEMENTS

Previous works (Yu et al., 2025; Liu et al., 2025b;d; An et al., 2025) have demonstrated that encouraging exploration during RL is critical. We observe that this factor becomes even more crucial in VLA RL. Manipulation tasks typically allow for a wide range of valid solutions. However, VLA models tend to converge on a narrow set of solution patterns, largely due to the homogeneity of their training trajectories, which limits the efficiency of RL. Promoting exploration encourages models to discover novel strategies and broaden the solution space, a property that is particularly advantageous in scenarios with low success rates. Building on this insight, we implement three key modifications to enhance the exploration of RL training: 1) employing dynamic sampling during trajectory rollout, 2) adjusting the clip range in the GRPO training objective, 3) and increasing the sampling temperature during rollout.

**Dynamic Sampling** Critic-free RL algorithms suffer from vanishing gradients when trajectories are assigned the same rewards. For example, GRPO computes advantages using group-relative normalization, comparing each response's reward to the mean and standard deviation of rewards within its group of sampled outputs. When all trajectories share identical rewards, their advantage estimation becomes zero, resulting in null gradients and causing unstable training dynamics.

We address this challenge through Dynamic Sampling (Yu et al., 2025; Cui et al., 2025a), a method that has been proven effective in LLM RL (Cui et al., 2025a; Yu et al., 2025; Team et al., 2025b; Shi et al., 2025). During rollout, we exclude groups in which all trajectories either succeed or fail. Sampling proceeds until the batch consists solely of groups with mixed outcomes, which can be formally expressed as:

$$0 < |\{\text{traj}_i(a_i, s_i) \mid \text{is\_successful}[\text{traj}_i(a_i, s_i)]\}| < G. \tag{2}$$

This ensures non-zero advantage estimates and stable gradient flow throughout training.

**Clipping Higher** PPO and GRPO employ clipping over the importance sampling ratio to restrict the trust region (Schulman et al., 2015) and enhance RL stability (Schulman et al., 2017a; Shao et al., 2024). However, the upper clipping threshold restricts the probability increase of low-probability tokens, thereby potentially constraining exploration. Following DAPO (Yu et al., 2025), we modify the clipping range in the GRPO training objective from [0.8, 1.2] to [0.8, 1.28].

**Higher Rollout Temperature** Recent works on LLM RL adjusting the rollout temperature to promote exploration have been widely shown to be effective, with sampling at higher temperatures yielding particularly notable improvements (An et al., 2025; Liu et al., 2025d; Liao et al., 2025). To encourage the VLA model to generate more diverse trajectories during the rollout phase, we increase the sampling temperature from 1.0 to 1.6. As shown in Figure 3, these modifications led to notable improvements.

## 2.4 TRAINING OBJECTIVE

We use the adopted GRPO algorithm (Shao et al., 2024) for online RL training on VLA models, with modifications as introduced in Section 2.3. Moreover, we remove the KL divergence regularization

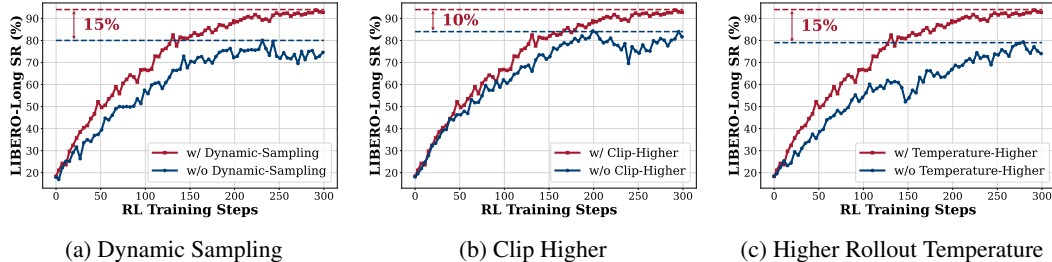

(a) Dynamic Sampling      (b) Clip Higher      (c) Higher Rollout Temperature

Figure 3: The effectiveness of three key enhancements: dynamic sampling, higher rollout temperature, and clip higher.

following DAPO (Yu et al., 2025). This eliminates the need for a reference model during training, reducing memory consumption and accelerating the training. Additionally, the KL penalty constrains policy divergence from a fixed reference, potentially limiting exploration of new behaviors. Therefore, the policy is optimized via the following objective:

$$\mathcal{J}(\theta) = \mathbb{E}_{s_0 \sim \mathcal{D}, \{a_t\}_{i=1}^{G} \sim \pi_{\theta_{\text{old}}}(\cdot | s_t)} \left[ \frac{1}{G} \sum_{i=1}^{G} \frac{1}{|a_i|} \sum_{t=1}^{|a_i|} \min \left( r_{i,t}(\theta)\hat{A}_i, \text{clip}\left( r_{i,t}(\theta), 1 - \varepsilon_L, 1 + \varepsilon_H \right) \hat{A}_i \right) \right]$$

$$\text{s.t.} \quad 0 < |\{\text{traj}_i(a_i, s_i) : \text{success}[\text{traj}_i(a_i, s_i)]\}| < G, \tag{3}$$

where

$$r_{i,t}(\theta) = \frac{\pi_\theta(a_{i,t} \mid s_{i,t})}{\pi_{\theta_{\text{old}}}(a_{i,t} \mid s_{i,t})}, \quad \hat{A}_i = \frac{R_i - \text{mean}(\{R_i\}_{i=1}^{G})}{\text{std}(\{R_i\}_{i=1}^{G})}. \tag{4}$$

Table 1: Main results of different VLA models on RoboTwin1.0.

| Model | RoboTwin1.0 | | | | |
|---|---|---|---|---|---|
| | Hammer Beat | Block Handover | Blocks Stack | Shoe Place | Avg |
| DP | 0.0 | 12.0 | 7.1 | 4.3 | 5.9 |
| DP3 | 64.7 | 84.3 | 24.0 | 59.3 | 58.1 |
| OpenVLA-OFT | 67.2 | 61.6 | 7.1 | 23.4 | 39.8 |
| **w/ ours** | **92.6** | **89.6** | **40.2** | **59.3** | **70.4** |
| Δ | +25.4 | +28.0 | +33.1 | +35.9 | +30.6 |

## 3 EXPERIMENTS

### 3.1 EXPERIMENTAL SETUP

**Benchmarks** We evaluate **SimpleVLA-RL** on three simulation benchmarks—LIBERO (Liu et al., 2023), RoboTwin1.0 (Mu et al., 2025), and RoboTwin2.0 (Chen et al., 2025a), and conduct real-world experiments on RoboTwin2.0 tasks. LIBERO is a language-guided manipulation benchmark with five task suites: LIBERO-Goal, LIBERO-Spatial, LIBERO-Object, LIBERO-Long (10 tasks each with 50 demonstrations), and LIBERO-90 (90 tasks for large-scale evaluation). Performance is measured by average Success Rate (SR) across 50 held-out test scenarios per task. RoboTwin1.0 provides 17 bimanual tasks, while RoboTwin2.0 extends to 50 tasks with 731 object instances and comprehensive domain randomization (clutter, lighting, background, tabletop height, language instructions), enhancing task diversity and sim-to-real transfer. For RoboTwin2.0, we use the Agilex Piper robotic arm and domain-randomized task settings, with each task evaluated on 100 held-out test scenarios. We select 12 tasks in RoboTwin2.0 and categorize them into 4 horizon levels based on average step counts, as detailed in Table 7.

**Backbones** We apply **SimpleVLA-RL** to OpenVLA-OFT (Kim et al., 2025), a state-of-the-art auto-regressive VLA model with high performance and inference efficiency. Built on Open-VLA (Kim et al., 2024), it uses vision encoders and LLaMA2-7B (Touvron et al., 2023) as the

Table 2: Main results of different VLA models on RoboTwin2.0, organized by task horizon.

| Short Horizon Tasks (100-130 Steps) | | | | | |
|---|---|---|---|---|---|
| **Model** | **Lift Pot** | **Beat Hammer Block** | **Pick Dual Bottles** | **Place Phone Stand** | **Avg** |
| $\pi_0$ | 51.0 | 59.0 | 50.0 | 22.0 | 45.5 |
| RDT | 45.0 | 22.0 | 18.0 | 13.0 | 24.5 |
| OpenVLA-OFT | 10.1 | 28.1 | 29.7 | 17.1 | 21.3 |
| **w/ ours** | **64.1** | **87.5** | **68.3** | **39.6** | **64.9** |
| $\Delta$ | +54.0 | +59.4 | +38.6 | +22.5 | +43.6 |
| Medium Horizon Tasks (150-230 Steps) | | | | | |
| **Model** | **Move Can Pot** | **Place A2B Left** | **Place Empty Cup** | **Handover Mic** | **Avg** |
| $\pi_0$ | 41.0 | 38.0 | 84.0 | 96.0 | 64.8 |
| RDT | 33.0 | 21.0 | 42.0 | 95.0 | 47.8 |
| OpenVLA-OFT | 28.1 | 37.5 | 77.3 | 45.3 | 47.1 |
| **w/ ours** | **61.2** | **45.3** | **94.2** | **89.2** | **72.5** |
| $\Delta$ | +33.1 | +7.8 | +16.9 | +43.9 | +25.4 |
| Long (280-320 Steps) & Extra Long Horizon Tasks (450-650 Steps) | | | | | |
| **Model** | **Handover Block** | **Stack Bowls Two** | **Blocks Rank Rgb** | **Put Bottles Dustbin** | **Avg** |
| $\pi_0$ | 39.0 | 53.0 | 45.0 | 54.0 | 47.8 |
| RDT | 26.0 | 42.0 | 17.0 | 26.0 | 27.8 |
| OpenVLA-OFT | 33.1 | 40.6 | 70.2 | 42.2 | 46.5 |
| **w/ ours** | **57.8** | **75.8** | **81.3** | **60.9** | **69.0** |
| $\Delta$ | +24.7 | +35.2 | +11.1 | +18.7 | +22.4 |
| **Overall Avg** | **RDT:** 33.3 | $\pi_0$**:** 52.7 | **OpenVLA-OFT:** 38.3 | **w/ ours:** 68.8 | +30.5 |

backbone with action chunk and parallel decoding, making it suitable for online RL's frequent inference requirements. Our implementation of the OpenVLA-OFT differs from the official version (see Appendix G.2 for modifications and Appendix G.3 for hyperparameters).

**Baselines** We compare with advanced VLA models: UniVLA (Bu et al., 2025b), RDT-1B (Liu et al., 2024), $\pi_0$ (Black et al., 2024), $\pi_{\text{fast}}$ (Pertsch et al., 2025), Nora (Hung et al., 2025), Open-VLA (Kim et al., 2024), Octo (Team et al., 2024), DP (Chi et al., 2024) and DP3 (Ze et al., 2024).

## 3.2 MAIN RESULTS

We evaluate **SimpleVLA-RL** on LIBERO, RoboTwin1.0, and RoboTwin2.0 using a two-stage paradigm: SFT followed by **SimpleVLA-RL** on OpenVLA-OFT, while baselines use SFT only. For LIBERO's four task suites, we perform SFT with 500 demonstrations per task suite, then RL on 500 simulation scenarios. For RoboTwin1.0, we use 50 demonstrations per task for single-task SFT, then 100 scenarios per task for RL. For RoboTwin2.0, we use 1,000 demonstrations per task for single-task SFT, then 1,000 scenarios per task for RL.

Tables 1, 2, and 3 present results on LIBERO, RoboTwin1.0, and RoboTwin2.0 benchmarks. On LIBERO, SimpleVLA-RL improves OpenVLA-OFT from 91% to 99% average success rate, achieving SoTA performance and surpassing models like $\pi_0$ and UniVLA. For long-horizon tasks in LIBERO-Long, SimpleVLA-RL reaches 98.5% success rate, with a 12% improvement over baseline and 13.3% over $\pi_0$. On RoboTwin1.0's dual-arm tasks, SimpleVLA-RL achieves 30.6% gains (39.8% to 70.4%). Across RoboTwin2.0's 12 tasks, SimpleVLA-RL delivers 80%

Table 3: Main results on LIBERO.

| **Model** | **LIBERO** | | | | |
|---|---|---|---|---|---|
| | **Spatial** | **Object** | **Goal** | **Long** | **Avg** |
| Octo | 78.9 | 85.7 | 84.6 | 51.1 | 75.1 |
| OpenVLA | 84.7 | 88.4 | 79.2 | 53.7 | 76.5 |
| Nora | 92.2 | 95.4 | 89.4 | 74.6 | 87.9 |
| $\pi_0$ + FAST | 96.4 | 96.8 | 88.6 | 60.2 | 85.5 |
| $\pi_0$ | 96.8 | 98.8 | 95.8 | 85.2 | 94.2 |
| UniVLA | 96.5 | 96.8 | 95.6 | 92.0 | 95.2 |
| OpenVLA-OFT | 91.6 | 95.3 | 90.6 | 86.5 | 91.0 |
| **w/ ours** | **99.4** | **99.1** | **99.2** | **98.5** | **99.1** |
| $\Delta$ | +7.8 | +3.8 | +8.6 | +12.0 | +8.1 |

relative improvement (38.3% to 68.8%), outperforming SoTA methods including $\pi_0$ (52.7%) and RDT (33.3%). Even on Extra-Long-Horizon tasks like "Blocks Rank Rgb" and "Put Bottles Dustbin", SimpleVLA-RL achieves 11.1% and 18.7% point gains respectively, demonstrating effectiveness across all horizon levels. These results validate that SimpleVLA-RL consistently

improves model performance across diverse benchmarks without requiring additional demonstration data, proving the effectiveness of outcome-level rewards even for complex long-horizon tasks.

# 4 ANALYSIS

In this section, we analyze the role of **SimpleVLA-RL** in addressing three key challenges that hinder the further advancement and scaling of the VLA model: **data, generalization, and real-world tasks**. Below are several key takeaways:

> **Takeaways**
>
> 1. **Data: SimpleVLA-RL** can significantly reduce reliance on demonstration data, effectively alleviating the data scarcity bottleneck that constrains VLA scaling (§ **4.1**).
> 2. **Generalization:** Compared to SFT, **SimpleVLA-RL** demonstrates strong generalization in spatial configurations, object types, and task settings (§ **4.2**).
> 3. **Real-world Task: SimpleVLA-RL** exhibits strong sim-to-real transfer, with large-scale simulation training remarkably improving real-world performance, indicating a promising path for scaling up real-world policy (§ **4.3**).

## 4.1 OVERCOMING DATA SCARCITY

Developing foundation VLA models for manipulation tasks requires large-scale demonstration data for training (Liu et al., 2024; Black et al., 2024; Intelligence et al., 2025). This data scaling paradigm has been proven in the NLP area (Hoffmann et al., 2022; Achiam et al., 2023; Touvron et al., 2023). However, acquiring high-quality trajectory data for embodied manipulation tasks remains expensive and difficult, creating a fundamental bottleneck for VLA model development (Bi et al., 2025; Zhong et al., 2025). Therefore, we investigate whether **SimpleVLA-RL** can enhance VLA models even with extremely limited demonstration trajectories to overcome this limitation.

**Settings** To simulate scenarios with scarce demonstration data, we finetune OpenVLA-OFT using only one demonstration data per task, denoted as *One-Trajectory SFT*. Given that each of the four LIBERO task suites contains 10 distinct tasks, we utilize merely 10 demonstration data per task suite. For comparison, we also conduct an experiment using all available demonstration data for each task, 500 per task suite, denoted as *Full-Trajectory SFT*. Following both *One-Trajectory SFT* and *Full-Trajectory SFT*, we apply **SimpleVLA-RL** on the SFT model.

**Results** As shown in Table 4, SFT performance degrades significantly with limited data. Under *One-Trajectory SFT*, success rates drop below 63.6% for LIBERO-Spatial/Object/Goal and to only 17.3% for LIBERO-Long, compared to 91.0% average under *Full-Trajectory SFT*. Remarkably, applying SimpleVLA-RL to *One-Trajectory SFT* models increases the average success rate from 48.9% to 96.9%, surpassing even *Full-Trajectory SFT*'s 91.0%. LIBERO-Long improves dramatically from 17.3% to 91.7%, while the other three task suites all

Table 4: Comparisons between One-Trajectory and Full-Trajectory SFT on LIBERO.

| Model | LIBERO | | | | |
|---|---|---|---|---|---|
| | **Spatial** | **Object** | **Goal** | **Long** | **Avg** |
| One-Trajectory SFT | | | | | |
| OpenVLA-OFT | 63.6 | 54.9 | 59.6 | 17.3 | 48.9 |
| w/ ours | **98.2** | **98.7** | **98.8** | **91.7** | **96.9** |
| Δ | +34.6 | +43.8 | +39.2 | +74.4 | +48.0 |
| Full-Trajectory SFT | | | | | |
| OpenVLA-OFT | 91.6 | 95.3 | 90.6 | 86.5 | 91.0 |
| w/ ours | **99.4** | **99.1** | **99.2** | **98.5** | **99.1** |
| Δ | +7.8 | +3.8 | +8.6 | +12.0 | +8.1 |

exceed 98%. The performance gap between *One-Trajectory SFT + RL* (96.9%) and *Full-Trajectory SFT + RL* (99.1%) is merely 2.2%. The results demonstrate that **SimpleVLA-RL** can substantially improve performance even in data-scarce scenarios, suggesting that online RL enables further scaling of VLA training through trial-and-error exploration, even with minimal demonstration data.

## 4.2 GENERALIZATION ANALYSIS

The generalization ability of VLA models remains a key challenge (Intelligence et al., 2025; Zhong et al., 2025; Liu et al., 2025a). This subsection evaluates how SFT and online RL methods like **SimpleVLA-RL** affect VLA generalization across three dimensions: spatial (LIBERO-Spatial), objects (LIBERO-Object), and tasks (LIBERO-Goal).

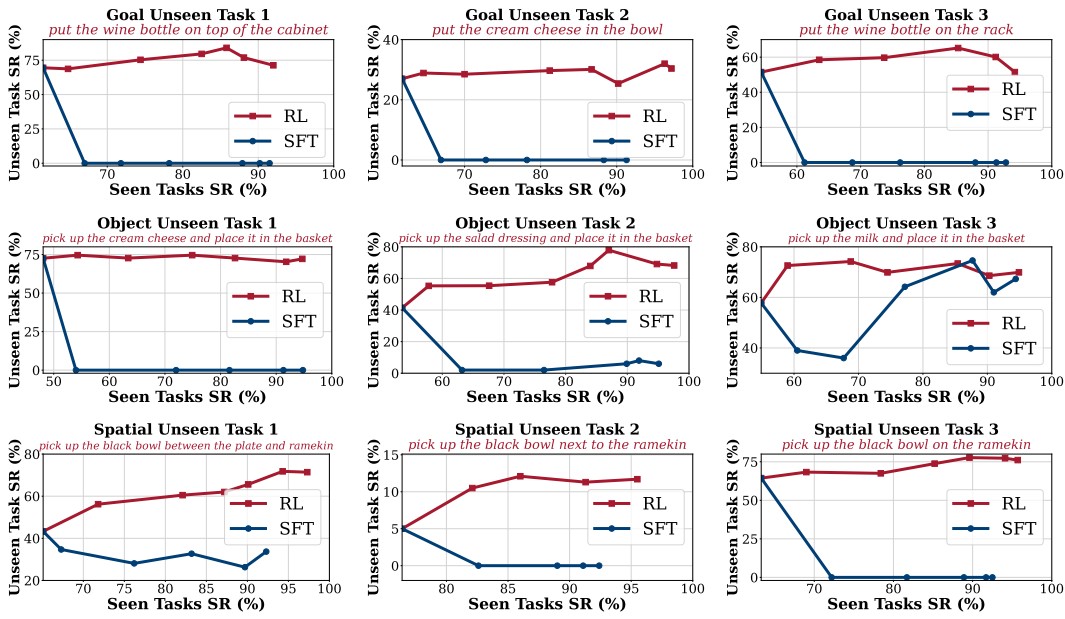

Figure 4: Generalization Analysis on LIBERO: Goal Unseen (**Top**), Object Unseen (**Middle**), Spatial Unseen (**Bottom**).

**Settings** We experiment on three LIBERO task suites (Spatial, Object, Goal), each containing ten tasks. For each suite, we randomly select nine tasks as seen tasks for RL or SFT training, while reserving the remaining task as the unseen task for out-of-distribution evaluation. For both methods, we first fine-tune OpenVLA-OFT under the *One-Trajectory SFT* setting to obtain a base model with non-zero success rates, since the original model achieves 0% on LIBERO and is incapable of performing online RL. For SFT, we further fine-tune the *One-Trajectory SFT* base model (§4.1) using 450 demonstrations from 9 seen tasks on each task suite. For RL, we perform **SimpleVLA-RL** on the *One-Trajectory SFT* base model using 450 scenarios from 9 seen tasks. We plot how unseen task performance evolves as training task success rates increase during training.

**Results** Figure 4 presents the results. While both SFT and RL achieve over 90% success rates on training tasks, their performance on unseen tasks diverges significantly. As training progresses, SimpleVLA-RL shows consistent improvement on unseen tasks across all settings, whereas SFT suffers from severe overfitting, often experiencing catastrophic forgetting with success rates of unseen tasks dropping to 0%. **On LIBERO-Goal**, SFT immediately drops to 0% on all three unseen tasks at training onset, likely because these tasks involve diverse objects and manipulation strategies with minimal transferable components. In contrast, SimpleVLA-RL maintains performance and achieves 5%-15% improvements. **On LIBERO-Object**, SFT improves only on Unseen Task 3 (57.8% to 74.6%) while failing on the other two. SimpleVLA-RL improves across all three tasks, with notable gains of 36.5% on Task 2 and 16.4% on Task 3. **On LIBERO-Spatial**, SFT degrades by 10% on Unseen Task 1 and completely fails on the remaining tasks, while SimpleVLA-RL improves Task 1 performance from 43.3% to 71.8% and achieves 7.1% and 13.3% gains on the other tasks. These results demonstrate that RL training enables VLA models to retain previously acquired capabilities while learning generalizable skills from diverse tasks.

### 4.3 REAL-WORLD EXPERIMENTS

Table 5: Real-world experiment (sim2real) results.

|  | Stack Bowls | Place Empty Cup | Pick Bottle | Click Bell | Avg |
|---|---|---|---|---|---|
| RDT | 60.0 | 4.0 | 10.0 | 20.0 | 23.5 |
| OpenVLA-OFT | 38.0 | 2.0 | 0.0 | 30.0 | 17.5 |
| **w/ ours** | 70.0 | 10.0 | 14.0 | 60.0 | 38.5 |
| Δ | +32.0 | +8.0 | +14.0 | +30.0 | +21.0 |

To evaluate the real-world effectiveness of **SimpleVLA-RL**, we conduct sim-to-real experiments on four RoboTwin2.0 tasks (detailed in Appendix G.1) : Stack Bowls, Handover Block, Pick Bottle, and Click Bell. We employ OpenVLA-OFT as the policy model, RDT as the baseline model, and execute on two AgileX Piper robotic arms. For each task, we first use 1000 simulation trajectories for SFT. Then we apply **SimpleVLA-RL** on the SFT model using 1000 simulation scenarios to obtain an RL model. The entire training process uses only simulation data without any real-world demonstrations. We evaluate both the SFT and RL models on clean tabletops with unseen backgrounds in the real world. Each task is tested with 50 trials. The RDT baseline model only undergoes the SFT stage.

The sim2real results in Table 5 demonstrate that **SimpleVLA-RL** significantly improves the real-world success rates of VLA models, with an average improvement from 17.5% to 38.5%, surpassing RDT's 23.5%. For instance, in the Stack Bowls task, **SimpleVLA-RL** achieves a 84% relative improvement, lifting performance from 38% to 70%.On the Pick Bottle task, which demands higher action precision, as the bottle will fall if the robotic arm is not perfectly aligned on the first attempt, the SFT model fails completely while **SimpleVLA-RL** achieves a 14% success rate. This demonstrates RL's effectiveness in improving action precision. Using **SimpleVLA-RL** for low-cost, large-scale, and highly parallel RL training in simulation, we significantly improve the real-world performance of simulation-trained VLA models. This demonstrates a promising path for scaling real-world policies: using rich simulation assets and high-fidelity simulators for cost-effective RL training to achieve superior performance in real-world deployment.

## 5 DISCUSSIONS

### 5.1 "PUSHCUT": EMERGENCE OF NEW PATTERNS THROUGH RL

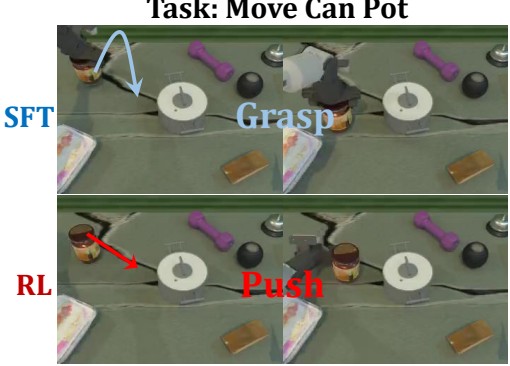
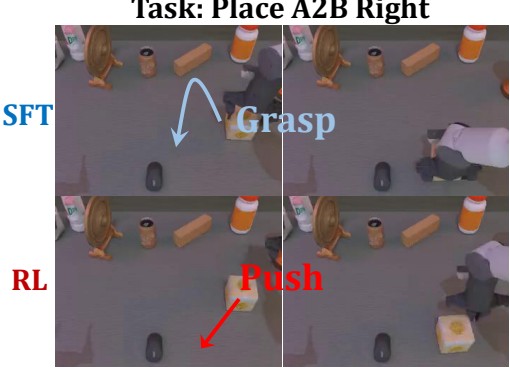

(a) "move can pot" task: Model learned to push the can to the pot (bottom) instead of grasp-move-place in the demonstration data (top).

(b) "place a2b right" task: Model learned to push A to B's right (bottom) instead of demonstrated grasp-move-place (top).

Figure 5: Illustration of "pushcut". Emergent pushing behaviors through RL in RoboTwin2.0 tasks.

During RL training with **SimpleVLA-RL**, we observe an emergent behavior we call "pushcut" (a push-driven shortcut), where the VLA model discovers novel strategies absent from the demonstration data. In the *move can pot* task of RoboTwin2.0, all demonstrations follow a grasp–move–place strategy (Figure 5a, top). However, after RL training, the model autonomously discovers a more efficient solution: directly pushing the can to the target location instead of grasping it (Figure 5a, bottom). Similar behaviors emerge in the *place a2b left/right* task, where the RL-trained model learns to push Object A into position rather than following the demonstrated grasp-move-place approach (Figure 5b).

This "pushcut" phenomenon parallels the "Aha Moment" in DeepSeek-R1 (Guo et al., 2025a), as both emerge through RL-driven exploration. This phenomenon highlights the fundamental distinction between SFT and RL. While SFT merely replicates patterns from demonstrations, RL enables the discovery of novel strategies through reward-driven exploration. The outcome-level reward design is crucial here: since both grasping and pushing receive equal rewards upon task completion, the sparse reward structure avoids procedural constraints, allowing the agent to explore freely and discover unexpected yet effective solutions.

## 5.2 FAILURE MODES OF SIMPLEVLA-RL

We conduct ablation studies to identify failure conditions and key influencing factors of **SimpleVLA-RL** (see Appendix C for full results and analysis). Our experiments reveal that model priors are the critical factor determining RL effectiveness. RL fails completely when the base model lacks initial task capability (0% success rate). Furthermore, we find a performance threshold: when initial success rates are too low ($< 5\%$), RL improvements remain negligible, while stronger initial models achieve substantially better final performance after RL training.

## 6 CONCLUSION

In this work, we present **SimpleVLA-RL**, an RL framework tailored for VLA models that extends veRL with VLA-specific trajectory sampling and parallelized training–inference–rendering capabilities. **SimpleVLA-RL** demonstrates significant improvements in data efficiency, generalization, and sim-to-real transfer. These results across LIBERO and RoboTwin benchmarks highlight RL's potential to both alleviate data scarcity and substantially enhance VLA generalization, paving the way for more autonomous and adaptable robotic models.

## ACKNOWLEDGEMENTS

This work is supported by National Science and Technology Major Project (2023ZD0121403), Young Elite Scientists Sponsorship Program by CAST (2023QNRC001), National Natural Science Foundation of China (No. 62406165), and Shanghai Municipal Science and Technology Major Project. We thank anonymous reviewers for their insightful comments and suggestions.

## ETHICS STATEMENT

This work presents **SimpleVLA-RL**, a reinforcement learning framework for improving Vision-Language-Action models in robotic manipulation. Our research aims to advance autonomous robotics for beneficial applications in manufacturing, healthcare, and assistive technologies. We conduct experiments exclusively in simulation environments and controlled laboratory settings with standard manipulation tasks, ensuring safe development practices. Our approach reduces reliance on large-scale human-operated data collection, minimizing both human labor costs and potential safety risks associated with extensive teleoperation. By enabling more sample-efficient training and better generalization, **SimpleVLA-RL** promotes environmentally conscious research through reduced computational requirements compared to scaling supervised learning alone. All experiments use publicly available benchmarks and models to ensure transparent, reproducible research.

## REPRODUCIBILITY STATEMENT

We provide comprehensive details to ensure reproducibility of our work. The complete algorithmic formulation of **SimpleVLA-RL** and training procedures are described in Section 2, including trajectory sampling, loss computation, and exploration strategies. All experimental configurations, model hyperparameters, hardware specifications, and robotic arm setups are detailed in Appendix G. We provide implementation specifics built upon the open-source veRL framework, evaluation protocols for LIBERO and RoboTwin benchmarks, and baseline comparisons. Additionally, we include ablation study configurations and real-world deployment settings. All mathematical formulations, implementation details, and experimental configurations necessary for reproducing our results are included in the paper.

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

# A    RELATED WORKS

## A.1    REINFORCEMENT LEARNING FOR LARGE LANGUAGE MODELS

Reinforcement Learning (RL) for Large Language Models (LLMs) has achieved remarkable success, demonstrating its ability to induce complex reasoning behaviors such as self-verification and iterative optimization, thereby significantly enhancing model performance on reasoning tasks (Guo et al., 2025a; Jaech et al., 2024; Liu et al., 2025c; Cui et al., 2025a; Zeng et al., 2025; Zuo et al., 2025). Recent advancements in Large Reasoning Models (LRMs), such as DeepSeek-R1 (Guo et al., 2025a), highlight the effectiveness of RL in boosting reasoning capabilities even with simple rule-based rewards, as exemplified by GRPO (Shao et al., 2024). This approach differs substantially from Reinforcement Learning from Human Feedback (RLHF) (Ouyang et al., 2022), which aligns base models with human preferences using algorithms like Proximal Policy Optimization (PPO) (Schulman et al., 2017a) and heavily relies on preference modeling.

Recent studies have increasingly focused on enhancing exploration in reinforcement learning to enable longer training horizons and improved performance. DAPO (Yu et al., 2025) introduces Clip-Higher, a decoupled variant of PPO clipping, which sets a higher upper bound relative to the lower one (e.g., $\varepsilon_L = 0.2$, $\varepsilon_H = 0.28$). This adjustment allows low-likelihood but potentially valuable tokens to increase in probability, thereby encouraging exploration. Building on this, POLARIS (An et al., 2025) employs a staged curriculum of temperature increases (e.g., $0.7 \rightarrow 1.0 \rightarrow 1.1$ for a 7B model) to gradually expand trajectory diversity and facilitate more robust policy discovery. In parallel, Entropy Mechanism (Cui et al., 2025b) addresses entropy collapse, a persistent issue in extended training, through methods such as Clip-Cov and KL-Cov, which selectively clip probabilities or penalize high-covariance tokens to sustain effective exploration. Similarly, ProRL (Liu et al., 2025b) combines KL control with reference policy resetting to preserve stability and extend training without degrading performance. A complementary line of work regulates entropy via temperature tuning. Acereason-nemotron 1.1 (Liu et al., 2025d) advocates adjusting temperatures to stabilize post-scaling entropy around a target (e.g., 0.3), balancing exploration and exploitation. Liao et al. (2025) further proposes a dynamic scheduler that adapts temperature over time to maintain stable entropy, thereby supporting sustained performance gains.

## A.2    VISION LANGUAGE ACTION MODELS

In the field of robotic manipulation tasks, VLA models (Kim et al., 2024; 2025; Liu et al., 2024; Bu et al., 2025b; Hung et al., 2025; Black et al., 2024; Pertsch et al., 2025; Intelligence et al., 2025) have shown better performance and task generalization compared to traditional policy-based approaches (Ma et al., 2022; Yuan et al., 2024). These models integrate the VLM or LLM backbone with action modules through unified end-to-end training (Zhong et al., 2025). This approach enables comprehensive multimodal understanding and fine-grained motor control (Firoozi et al., 2025). Currently, many studies are focused on enhancing the effectiveness of VLA models. For example, E-COT (Zawalski et al., 2024; Chen et al., 2025b) introduced Embedded Chain of Thought (ECoT) to improve the spatial reasoning ability of VLA models. RDT-1B and VPP (Liu et al., 2024; Hu et al., 2024) proposed diffusion-based frameworks for VLA models. Agibot world and Roboverse (Geng et al., 2025; Bu et al., 2025a) aim to build larger-scale simulation environments and trajectory datasets to improve the sim-to-real transfer and generalization capabilities of VLA models. Additionally, Dexmimicgen (Jiang et al., 2024) explores automated methods to generate high-quality trajectory data to address the issue of data scarcity in robotics. Despite the rapid advancements in the VLA domain, imitation learning remains the dominant training paradigm for VLA models (Sapkota et al., 2025; Kim et al., 2024; 2025; Liu et al., 2024; Bu et al., 2025b; Hung et al., 2025; Black et al., 2024; Pertsch et al., 2025; Intelligence et al., 2025). Current VLA models typically follow a two-stage paradigm: pretraining on multimodal data (e.g., Open X-Embodiment (O'Neill et al., 2024)) followed by SFT on collected robot trajectories. However, imitation learning is limited by its dependence on expensive, high-quality trajectory data and poor generalization to unseen scenarios.

**VLA RL Methods**    Recently, some efforts have attempted to apply RL to VLA training. GRAPE (Zhang et al., 2024) utilized Direct Preference Optimization (DPO) (Rafailov et al., 2023) to train VLA models by integrating human preferences. ConRFT (Chen et al., 2025c) introduced Reinforced Fine-Tuning (Trung et al., 2024) to train VLA models in real-world environments, iteratively training VLAs through alternating RL and SFT rounds. ReinboT (Zhang et al., 2025) fo-

cused on dense reward design and optimized VLA models through reward maximization. Guo et al. (2025b) proposed an iterative training framework that combines Supervised Fine-Tuning (SFT) and RL stages to address training instability and computational overhead. More recent works have further advanced VLA RL methods. Concurrently, RIPT-VLA (Tan et al., 2025) investigates a closely related problem, employing RLOO (Ahmadian et al., 2024) for VLA RL training. Moreover, Liu et al. (2025a) investigates RL's impact on VLA generalization capabilities, demonstrating significant improvements over SFT in unseen environments, objects, and textures. RLinf (Team, 2025) designed a flexible, scalable framework for VLA RL that unifies rendering, inference, and training, improving both VLA training efficiency and performance. VLA-RL (Lu et al., 2025) applies the PPO algorithm to the VLA model. TGRPO (Chen et al., 2025d) uses Claude3.7 to evaluate trajectories and optimizes VLA with GRPO. RFTF (Shu et al., 2025) uses value models to generate dense rewards in embodied scenarios for VLA online RL. Compared to the above works, our paper further explores the effectiveness of VLA RL on real-world robotic tasks. We also conduct comprehensive analyses on how VLA RL addresses data scarcity challenges and improves policy generalization.

## B    Preliminaries

To provide an intuitive illustration of the existing gap when extending RL methodologies from LLMs to the VLA domain, we formalize RL for both LLMs and VLA models, presenting their state representations, action spaces, reward functions, and environments in this section.

### B.1    RL Formulation for LLMs

**State ($s_t$):** At step $t$, the state $s_t$ comprises the input prompt and previously generated tokens:

$$s_t = (x_{\text{prompt}}, y_1, y_2, \ldots, y_{t-1}),  \tag{5}$$

where $x_{\text{prompt}}$ denotes the initial prompt and $y_t$ denotes the $t$-th generated token.

**Action ($a_t$):** An action corresponds to selecting the next token from the vocabulary $\mathcal{V}$. At each step, the policy outputs a probability distribution over tokens, and the action token is selected via random sampling. Formally, the action is defined as:

$$a_t = y_t \in \mathcal{V}, \quad \text{where} \quad y_t \sim \pi_\theta(\cdot|s_t) = \text{softmax}\left(f_\theta(s_t)/T\right),  \tag{6}$$

where $f_\theta(s_t) \in \mathbb{R}^{|\mathcal{V}|}$ represents the LLM logit outputs and $T$ is the temperature parameter controlling the randomness of sampling.

**Environment:** The environment provides reward signals upon sequence completion. In rule-based settings, binary rewards are assigned based on the correctness. Alternatively, learned reward models or human feedback systems provide continuous rewards based on criteria such as helpfulness, harmlessness, or task alignment. The reward is computed as follows:

$$r(\tau) = \begin{cases} 1, & \text{if } \tau \text{ satisfies correctness criteria} \\ 0, & \text{otherwise} \end{cases}, \quad \text{or} \quad r(\tau) = R_\phi(\tau) \in [0, 1],  \tag{7}$$

where $R_\phi$ is a learned reward model and $\tau = (x_{\text{prompt}}, y_1, y_2, \ldots, y_{T_{\text{seq}}})$ represents the complete generated sequence of length $T_{\text{seq}}$.

**Rollout:** Given an input prompt $x_{\text{prompt}}$, the LLM auto-regressively generates a sequence by sampling tokens from $\pi_\theta(y_t|s_t)$ until termination, without intermediate environmental feedback. With a non-zero temperature $T$, the policy can produce diverse rollouts that explore different solution paths.

### B.2    RL Formulation for VLAs

**State ($s_t$):** The state consists of multimodal observations including visual input (RGB images, depth maps, or point clouds), proprioceptive information (joint angles, end-effector pose), and language instructions of the tasks. Formally, the state is defined as:

$$s_t = (o_t^{\text{vis}}, o_t^{\text{prop}}, l_{\text{task}}),  \tag{8}$$

where $o_t^{\text{vis}}$ is multimodal observations, $o_t^{\text{prop}}$ is proprioceptive information, and $l_{\text{task}}$ is language instructions.

**Action** ($a_t$)**:** Actions are control commands in the robot action space, typically end-effector deltas or joint angle targets, where $a_t \in \mathbb{R}^d$ (e.g., $d = 7$ for 6-DoF pose plus gripper position). Most VLA policies generate actions through either a diffusion-based action expert or a discrete action tokenizer. The action is defined as follows:

$$a_t = Decoder(h_\theta(s_t)), \quad Decoder \in \{\text{Diffusion Expert, Action Tokenizer}\}, \quad a_t \in \mathbb{R}^d, \quad (9)$$

where $h_\theta(s_t)$ represents the hidden state of $s_t$ in the VLA model, and $Decoder$ is the action decoder.

**Environment:** The environment represents the physical world or simulation where the robot operates. It provides state transitions $s_{t+1} = \text{Env}(s_t, a_t)$ and reward signals:

$$r_t = \alpha \cdot I_{\text{success}} + (1-\alpha) \cdot \sum_i w_i \cdot \phi_i(s_t, a_t), \quad \alpha \in [0, 1], \quad I_{\text{success}} = \begin{cases} 1, & \text{if task success} \\ 0, & \text{otherwise} \end{cases}, \quad (10)$$

where $\phi_i(s_t, a_t)$ represents process rewards (e.g. distance to goal), $w_i$ are weights, and $\alpha$ balances outcome and process rewards.

**Rollout:** VLA models generate trajectories through iterative interaction with the environment. At each timestep, the policy $\pi_\theta$ takes the current state $s_t$ as input and outputs an action chunk $(a_t, a_{t+1}, \ldots, a_{t+k-1})$ of length $k$. The robot executes these actions sequentially and the environment produces updated states based on physical dynamics. After execution, the model takes the new state $s_{t+k}$ as input and generates the next action chunk. This process continues until task completion or maximum episode length, producing a complete trajectory $\tau = ((s_0, a_0), (s_1, a_1), \ldots, (s_T, a_T))$ through interactive sampling.

### B.3 GROUP RELATIVE POLICY OPTIMIZATION

Group Relative Policy Optimization (GRPO) (Shao et al., 2024) is an RL method that eliminates the value function by computing advantages through group-relative normalization. Given an initial state $s_0$, the behavior policy $\pi_{\theta_{\text{old}}}$ generates $G$ trajectories $\{\tau_i\}_{i=1}^G$. The GRPO objective employs PPO-style clipping with KL regularization to constrain policy updates:

$$J_{\text{GRPO}}(\theta) = \mathbb{E}_{s_0 \sim \mathcal{D}, \{\tau_i\} \sim \pi_{\theta_{\text{old}}}} \left[ \frac{1}{G} \sum_{i=1}^G \frac{1}{|\tau_i|} \sum_{t=1}^{|\tau_i|} \min\left( r_{i,t}(\theta)\hat{A}_i, \text{clip}(r_{i,t}(\theta), 1-\epsilon, 1+\epsilon)\hat{A}_i \right) \right.$$
$$\left. - \beta D_{\text{KL}}(\pi_\theta || \pi_{\text{ref}}) \right], \quad (11)$$

where the importance sampling ratio $r_{i,t}(\theta)$ and the normalized advantage $\hat{A}_i$ are defined as:

$$r_{i,t}(\theta) = \frac{\pi_\theta(a_{i,t}|s_{i,t})}{\pi_{\theta_{\text{old}}}(a_{i,t}|s_{i,t})}, \quad \hat{A}_i = \frac{R_i - \text{mean}(\{R_i\}_{i=1}^G)}{\text{std}(\{R_i\}_{i=1}^G)}. \quad (12)$$

Here $R_i$ denotes the total reward of the $i$-th trajectory, $\epsilon > 0$ is the PPO clipping parameter that limits the policy ratio, and $\beta > 0$ is the coefficient controlling the strength of KL regularization with respect to the reference policy $\pi_{\text{ref}}$.

## C FAILURE MODES OF SIMPLEVLA-RL

This subsection investigates the failure conditions of **SimpleVLA-RL** and key influencing factors. Through experiments on five RoboTwin2.0 tasks, we find that the model priors are the critical factor determining RL effectiveness.

**Settings** Each task is trained under domain randomization with a single-task setting. We compare three model variants: (1) the OpenVLA-OFT base model without trajectory fine-tuning (0 trajectories SFT); (2) the model fine-tuned with 100 demonstration trajectories per task (100 trajectories SFT); and (3) the model fine-tuned with 1000 demonstration trajectories per task(1000 trajectories SFT). All models undergo **SimpleVLA-RL** training on 1000 training scenarios and are evaluated on 100 held-out test scenarios.

Table 6: Impact of initial model capability on **SimpleVLA-RL** performance.

| | RoboTwin2.0 | | | | | |
|---|---|---|---|---|---|---|
| | **Move Can Pot** | **Place A2B Left** | **Place A2B Right** | **Place Phone Stand** | **Pick Dual Bottles** | **Avg** |
| 0 trajs SFT | 0 | 0 | 0 | 0 | 0 | 0 |
| +RL | 0 | 0 | 0 | 0 | 0 | 0 |
| 100 trajs SFT | 9.4 | 7.8 | 7.8 | 10.1 | 1.2 | 7.3 |
| +RL | 51.6 | 25.0 | 27.2 | 18.8 | 4.3 | 25.4 |
| Δ | +42.2 | +17.2 | +19.4 | +8.7 | +3.1 | +18.1 |
| 1000 trajs SFT | 28.1 | 37.5 | 28.7 | 17.1 | 29.7 | 28.2 |
| +RL | 61.2 | 45.3 | 37.5 | 39.6 | 68.3 | 50.4 |
| Δ | +33.1 | +7.8 | +8.8 | +22.5 | +38.6 | +22.2 |

**RL fails completely when the base model has no initial task ability.** Table 6 reports the results. The base model (0-trajectory SFT) achieves a 0% success rate across all tasks, exhibiting no task-relevant behaviors. Despite extensive pretraining, OpenVLA shows extremely limited zero-shot generalization, consistent with findings in Kim et al. (2025). Because no successful trajectories are generated during sampling and only outcome rewards (without process rewards) are employed, every trajectory receives zero reward. As a result, RL is unable to improve performance, which remains at 0%.

**The model prior has a significant impact on the effectiveness of RL.** Initial capability is strongly correlated with post-RL performance. The 100-trajectory SFT model improves from 7.3% to 25.4% (an 18.1% gain), while the 1000-trajectory SFT model improves from 28.2% to 50.4% (a 22.2% gain) in average success rate. This trend is consistent across tasks. For instance, in the *move can pot* task, the 100-trajectory SFT model improves from 9.4% to 51.6%, whereas the 1000-trajectory SFT model improves from 28.1% to 61.2%. These results highlight that stronger initial capabilities provide more effective starting points for exploration, thereby facilitating greater performance improvements.

**RL effectiveness has a threshold: when initial ability is too low, improvements remain negligible.** Our findings further reveal that the effectiveness of RL is subject to a performance threshold. When initial success rates are very low, online RL with outcome rewards yields only marginal improvements. For example, in the *pick dual bottles* task, the 100-trajectory SFT model improves from 1.2% to 4.3%, while the 1000-trajectory SFT model improves from 29.7% to 68.3%. Similarly, in the *place phone* task, the 100-trajectory SFT model gains 8.7%, compared to a 22.5% gain for the 1000-trajectory SFT model. The results indicate that a minimal level of task competence is essential for effective RL. Below this threshold, exploration is ineffective and RL fails to produce meaningful gains.

# D    ADDITIONAL ABLATION AND ANALYSIS EXPERIMENTS

## D.1    ABLATION STUDY ON KL REGULARIZATION

In this section, we present the ablation study on KL regularization and analyze the training stability after removing the KL constraint. The experimental settings are consistent with Figure 3. We apply SimpleVLA-RL with OpenVLA-OFT on LIBERO-Long under the one-trajectory setting, comparing two configurations: with and without the KL constraint. The results are shown in Figure 6.To further verify the stability of model updates, we visualize the gradient norm, policy gradient loss, and sampling accuracy curves throughout training with and without the KL constraint in Figure 7.

The experimental results demonstrate that removing the KL regularization leads to a slight improvement in model performance while maintaining comparable training stability. The loss curves, gradient norms, and sampling accuracy all exhibit smooth convergence patterns. Furthermore, removing the KL constraint simplifies the training framework by eliminating the need to compute reference model sampling probabilities and load additional reference models, resulting in approximately 10% reduction in training time.

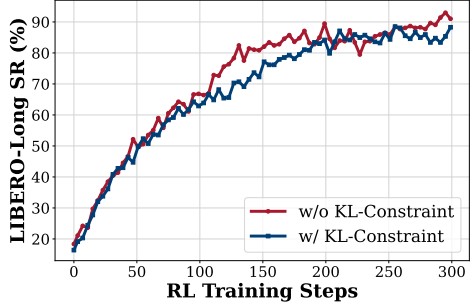

Figure 6: Ablation study on KL constraint on LIBERO-Long. Removing the KL constraint achieves comparable performance.

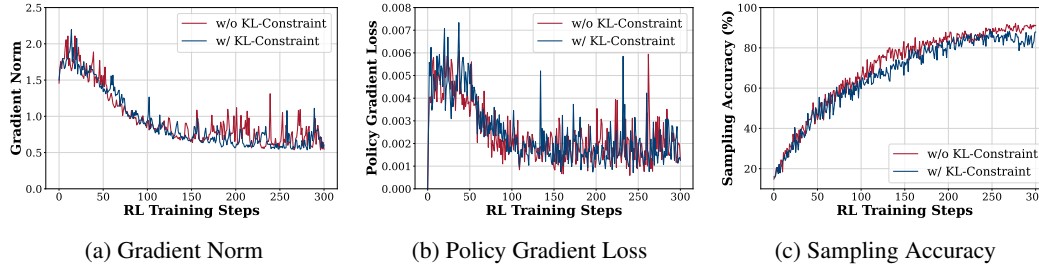

(a) Gradient Norm          (b) Policy Gradient Loss          (c) Sampling Accuracy

Figure 7: Training stability analysis with and without KL constraint on LIBERO-Long. Training without KL constraint remains equally stable.

## D.2  TRAINING STABILITY

We provide training curves for OpenVLA-OFT and Pi0.5 on the RoboTwin2.0 benchmark without KL regularization, including policy gradient loss, gradient norm, and sampling accuracy throughout training. Specifically, OpenVLA-OFT is trained on the Lift Pot task from RoboTwin2.0, as shown in Figure 8. Pi0.5 is trained on a mixture of 8 tasks from RoboTwin2.0: Lift Pot, Beat Hammer Block, Pick Dual Bottles, Place Phone Stand, Move Can Pot, Place A2B Left, Place Empty Cup, and Handover Mic, as shown in Figure 9. The results demonstrate stable training dynamics throughout the optimization process, with no significant fluctuations or instabilities observed in any of the metrics.

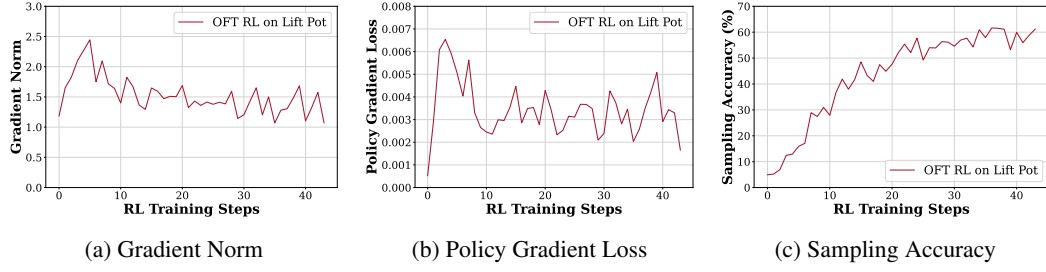

(a) Gradient Norm          (b) Policy Gradient Loss          (c) Sampling Accuracy

Figure 8: Training stability of OpenVLA-OFT on RoboTwin2.0 Lift Pot task. The gradient norm and policy gradient loss remain stable throughout training.

## E  MULTI-TASK RL TRAINING ON LIBERO-90

To evaluate the effectiveness of **SimpleVLA-RL** in multi-task mixed training settings, we conduct RL training on the LIBERO-90 task suite, which contains 90 different tasks. Each task consists of 50 scenarios, resulting in a total of 4,500 scenarios across all 90 tasks that serve as the training set for **SimpleVLA-RL**. Figure 10 shows the results of different models on LIBERO-90, as well as the performance of OpenVLA-OFT SFT and OpenVLA-OFT RL on LIBERO-90 under the one-trajectory SFT setting.

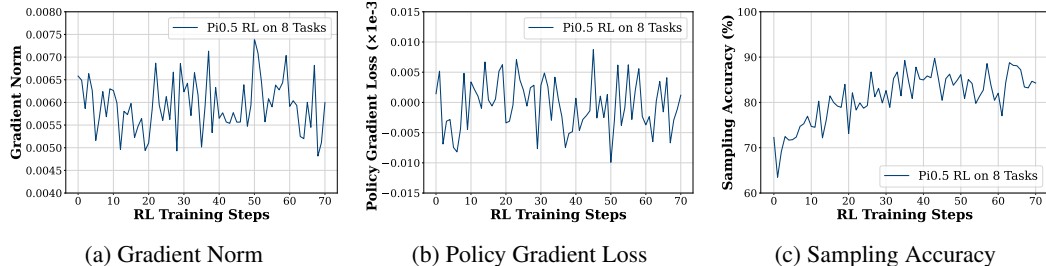

(a) Gradient Norm      (b) Policy Gradient Loss      (c) Sampling Accuracy

Figure 9: Training stability of Pi0.5 on RoboTwin2.0 with 8 tasks. The training metrics show stable optimization.

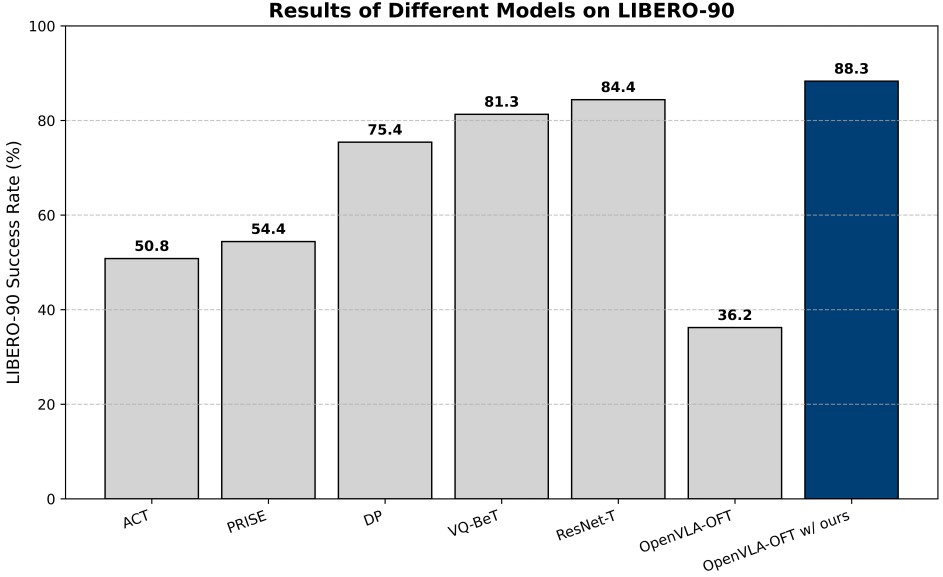

Figure 10: Results of different models on LIBERO-90.

## F    COMPARISON OF LLM AND VLA ROLLOUT ALGORITHMS

To better illustrate the concrete rollout process in **SimpleVLA-RL** and highlight the key differences between VLA rollout in **SimpleVLA-RL** and LLM rollout in the veRL framework, we present a comparative pseudo-code implementation in Listing 1.

```
def rollout(policy, dataset, number_sample=8, max_steps=None):
  rollout_dataset = []
  for batch in dataset:
    batch = batch.repeat(number_sample)
-    # LLM generates diverse outputs using random sampling
-    outputs = policy.generate(batch, temperature=1.0)
-    rollout_dataset.append((batch, outputs))
+    # Parallel env initialization and interaction
+    envs = env_process_pool.submit(batch.initialize)
+    states = env_process_pool.submit(envs.setup)
+    for t in range(max_steps):
+      # VLA generates diverse trajectories using temperature
  sampling on action tokens
+      actions = policy.generate(states, temperature=1.0)
+      rollout_dataset.append({f"{e.name}_step_{t}": (s,a) for e,s,a
  in zip(envs,states,actions)})
+      states, dones = env_process_pool.submit(envs.step, actions)
+      # Remove completed tasks
+      active = [(e,s) for e,s,d in zip(envs,states,dones) if not d]
+      if not active:
+        break
+      envs, states = zip(*active)
  return rollout_dataset
```

Listing 1: Pseudo-code for the adopted veRL rollout function: from LLM-based generation to interactive VLA sampling with synchronous environment parallelism.

## G    EXPERIMENTAL CONFIGURATION AND IMPLEMENTATION

### G.1    ROBOTWIN2.0 TASK CLASSIFICATION AND DETAILS

We classified the 12 tasks in RoboTwin2.0 based on their average number of steps, categorizing them into Short Horizon Tasks, Medium Horizon Tasks, Long Horizon Tasks, and Extra Long Horizon Tasks. The Table7 shows the specific number of steps and classification for each task.

Detailed descriptions of the 4 real-world tasks (Stack Bowls, Handover Block, Pick Bottle, and Click Bell) and 12 RoboTwin2.0 tasks can be found at `https://robotwin-platform.github.io/doc/tasks/index.html`.

### G.2    BACKBONE MODIFICATION DETAILS

Our implementation of the OpenVLA-OFT model differs from the official version. To achieve improved training and inference efficiency, we utilize only single-view images, language instructions, and robot proprioceptive states as model inputs, whereas the official model additionally incorporates wrist camera images. Additionally, in the LIBERO, we don't use robot proprioceptive states in model inputs. Regarding the model architecture, we employ only parallel decoding and action chunking designs. We use the LLaMA2 output head to generate action tokens and the cross-entropy loss, whereas the official model uses an MLP to generate continuous actions and L1 regression. Due to the differences in model inputs and architecture, we cannot use the official checkpoints. We modify the official codebase and performed SFT from scratch using the same datasets and hyperparameters as the official implementation.

Table 7: RoboTwin 2.0 task classification based on planning horizon and required steps.

| Task Name | Steps | Horizon | Horizon Group |
|---|---|---|---|
| **Short Horizon Tasks (112-130 steps)** | | | |
| lift_pot | 112 | Short | |
| beat_block_hammer | 113 | Short | Average: 121 steps |
| pick_dual_bottles | 127 | Short | Count: 4 tasks |
| place_phone_stand | 130 | Short | |
| **Medium Horizon Tasks (151-223 steps)** | | | |
| move_can_pot | 151 | Medium | |
| place_a2b_left | 155 | Medium | Average: 176 steps |
| place_empty_cup | 174 | Medium | Count: 4 tasks |
| handover_mic | 223 | Medium | |
| **Long Horizon Tasks (283-313 steps)** | | | |
| handover_block | 283 | Long | Average: 298 steps |
| stack_bowls_two | 313 | Long | Count: 2 tasks |
| **Extra Long Horizon Tasks (466-637 steps)** | | | |
| blocks_rank_rgb | 466 | Extra-Long | Average: 552 steps |
| put_bottles_dustbin | 637 | Extra-Long | Count: 2 tasks |
| **Overall Statistics** | **Total: 12 tasks, Average: 256 steps** | | |

### G.3 IMPLEMENTATION DETAILS AND HYPERPARAMETERS

For training infrastructure, we employ $8\times$NVIDIA A800 80GB for full-parameter training. The training hyperparameters are configured as follows: learning rate $lr = 5 \times 10^{-6}$, training batch size of 64, sampling count of 8, mini-batch size of 128, clip ratio $\varepsilon_L = 0.2$, $\varepsilon_H = 0.28$, and temperature $T = 1.6$. The number of action chunks is 8 in the LIBERO and 25 in the RoboTwin1.0&2.0. The model is configured with a total of 256 action tokens. The maximum interaction step is set to 512 in the LIBERO and 200, 400, or 800 in the RoboTwin1.0&2.0, depending on different tasks.

Regarding training time, the wall-clock training time varies by benchmark:

- **RoboTwin benchmark**: Using 8 A800-80GB GPUs, the wall-clock training time for single-task RL using OpenVLA-OFT is approximately 12-24 hours, depending on the initial success rate and maximum episode length. Training time increases when the initial success rate is lower and tasks require more steps.

- **LIBERO benchmark**: Under the *One-Trajectory SFT* setting, RL training takes approximately 1 or 2 days; under the *Full-Trajectory SFT* setting, training time is shorter, around 1 day. Tasks with lower initial success rates require longer training time.

RoboTwin benchmark: Using 8 A800 GPUs, the wall-clock training time for OpenVLA-OFT single-task RL is approximately 12-24 hours, depending on the task's initial success rate and maximum episode length. Training time increases when the initial success rate is lower or when tasks require more steps.

LIBERO benchmark: Under the one-traj setting, RL training takes longer, approximately 1-2 days; under the traj-all setting, training time is relatively shorter, around 1 day. Training time increases correspondingly when the initial success rate is lower.

During the rollout phase of RL training, we employ random sampling. For evaluation, we utilize greedy sampling, with each benchmark tested three times for reproducibility.

### G.4 ROBOT HARDWARE DETAILS

For real-world experiments, we employ an AgileX Cobot Magic, which is a mobile platform with an Aloha configuration consisting of four robotic arms. Each arm is an AgileX Piper with six degrees of freedom, equipped with a one-DoF parallel gripper. A RealSense D435 RGB-D camera is mounted

on the platform, capturing RGB images in real time at a resolution of $640 \times 480$ with a frame rate of approximately 30 Hz.

## H  THE USE OF LARGE LANGUAGE MODELS

We utilized LLMs for grammatical refinement and clarity improvements in our manuscript. Specifically, we used ChatGPT (GPT-5-Thinking) to help polish the language and correct grammatical errors in our draft. The assistance was limited to improving readability and ensuring adherence to academic writing conventions, while all technical content, experimental design, and scientific contributions remain entirely our own work.

