# OpenReview forum: "SimpleVLA-RL: Scaling VLA Training via Reinforcement Learning"
_ICLR.cc/2026/Conference — ICLR 2026 Poster_

### Official Review · Reviewer_urcu · 2025-10-29

**Soundness:** 4
**Presentation:** 4
**Contribution:** 4
**Rating:** 8
**Confidence:** 5

**Summary:**

This paper proposes SimpleVLA-RL, a practical online RL framework for Vision-Language-Action (VLA) models that scales beyond limited demonstration data. Building on veRL, the authors implement VLA-specific interactive rollout (closed-loop env stepping with token-based action distributions), simple outcome-only (0/1) trajectory rewards, and GRPO training with three exploration enhancers: dynamic sampling (discard all-success/all-failure groups), higher clipping range (0.8–1.28), and higher rollout temperature (1.6). They also drop the KL penalty (no reference model) to reduce memory and encourage exploration. Experiments fine-tune OpenVLA-OFT and then apply online RL across LIBERO, RoboTwin-1.0, RoboTwin-2.0, plus real-robot sim2real tests. Results show ~99% avg SR on LIBERO, large gains on RoboTwin (e.g., 38.3→68.8% overall on RT-2.0; 39.8→70.4% on RT-1.0), strong data-efficiency (with one demonstration per task: 48.9→96.9% on LIBERO), improved generalization to unseen spatial/object/goal tasks, and notable sim2real improvements.

**Strengths:**

1. Clear, simple RL recipe for VLAs (outcome-only rewards + GRPO) that removes reward-engineering and reference-model overhead; the method details (interactive rollout, reward assignment, GRPO objective) are explicit and easy to reproduce.

2. Exploration enhancements are well-motivated and ablated (dynamic sampling / higher clip / higher temperature), each contributing sizable gains.

3. Strong, broad empirical gains: SOTA-level LIBERO (~99% avg), large improvements across short/medium/long/extra-long horizons on RoboTwin-2.0, and substantial jumps on RoboTwin-1.0.

4. Data efficiency: with one traj per task, RL lifts LIBERO-Long from 17.3%→91.7% and overall 48.9%→96.9%, nearly closing the gap to full-data training.

5. Generalization: RL improves OOD tasks while SFT overfits/forgets—documented across spatial/object/goal suites with learning-progress plots.

**Weaknesses:**

(1) pic 5.1 is a bit confusing for me. I am not sure why the upper half has the different background to the bottom half. I think the best way to show emergent behavior is comparing behaviors under same conditions.

(2) Compute/throughput transparency: The paper lacks precise reporting of wall-clock training time, GPU type.

(3) Ablations indicate RL fails when the base success rate is ~0% and remains weak <5%; this limits plug-and-play use in very hard tasks without warm-starts.

**Questions:**

please see the weakness and try to answer them, especially weakness 3. I am curious about if you explored how to deal with it.

---

> ### Author Response · Authors · 2025-11-21
>
> ## W1: Emergent Behavior Under Same Conditions
>
> Thank you for pointing this out. Comparing emergent behaviors under the same scene and conditions is indeed more appropriate. The inconsistent backgrounds between the upper and lower scenes in Figure 5.1 was an oversight on our part. We have revised the figure (the updated figure can be found in Section 5.1 of the paper) to ensure the two scenes are consistent and properly aligned.
>
> ---
>
> ## W2: Compute/Throughput Transparency
>
> Thank you for pointing this out. We have already specified the GPU devices used (8×A800-80GB) and various training hyperparameters in Appendix G.3 of the original manuscript. In Appendix G.4, we also provided detailed information about the robot hardware and control frequency for real-robot experiments.
>
> **Training Time Information**:
>
> Regarding actual training time, we acknowledge this was missing from Appendix G.3. Thank you for the suggestion, we have now added the relevant training time information to Appendix G.3. Specifically:
>
> **RoboTwin Benchmark**:
> Using 8 A800 GPUs, the wall-clock training time for single-task RL with OpenVLA-OFT is approximately 12-24 hours, depending on the task's initial success rate and maximum episode length. Lower initial success rates or longer episode lengths result in increased training time.
>
> **LIBERO Benchmark**:
> Under the One-Trajectory Setting, RL training takes longer, approximately 1-2 day. Under the Full-Trajectory Setting, training time is relatively shorter, approximately 1 day. Lower initial success rates correspond to longer training times.

---

> ### Author Response · Authors · 2025-11-21
>
> ## W3 & Question: Plug-and-Play for Low Success Rate Tasks
>
> Thank you for raising this valuable point. We have mentioned in Section 5.2 that when the initial success rate of a task is extremely low or zero, performance improvement is limited or even unattainable. **This limitation is actually an inherent characteristic of RL itself**. RL requires the model to have a reasonably good prior to achieve meaningful improvements, and this requirement becomes more pronounced when using sparse rewards. Shunyu Yao's "The Second Half" blog [1] presents a similar perspective.
>
> For example, when the model lacks sufficient prior knowledge and cannot sample any successful trajectories, it fails to obtain any positive reward signals. Without these signals, the model cannot identify which trajectories should be reinforced, making effective training infeasible. This characteristic has also been verified in current LLM RL. For example, when performing RL on weaker LLMs with Olympiad-level difficulty problems, the model cannot sample successful trajectories and thus cannot learn effectively.
> However, we also hope to address your concerns with the following points:
>
>
> ### **(1) Acquiring Sufficient Prior for Effective RL Often Requires Minimal Data**
>
> In Section 4.1 (Overcoming Data Scarcity), we demonstrated that a single demonstration trajectory for SFT is sufficient to establish the necessary prior for effective RL on LIBERO. While this does not completely eliminate the dependence on demonstration data, it enables cold-start at very low cost, after which RL can drive the model to strong performance.
>
> ### **(2) Stronger Models Enable True Plug-and-Play Applicability**
>
> The OpenVLA-OFT we currently use is still a relatively weak model. However, with stronger models such as π0.5 [2], we observe that the model exhibits reasonable zero-shot capabilities. This means that when applying SimpleVLA-RL to new tasks not seen during training, we do not need to collect task-specific demonstration data for cold-start, the model possesses sufficient prior knowledge to enable direct RL training, achieving true plug-and-play applicability.
>
> ### **Experimental Validation**
>
> We conducted an experiment to support this point. We first performed multi-tasks SFT on the π0.5 base model using 8 tasks from RoboTwin 2.0 (*Lift Pot*, *Beat Hammer Block*, *Pick Dual Bottles*, *Place Phone Stand*, *Move Can Pot*, *Place A2B Left*, *Place Empty Cup*, *Handover Mic*), with 50 trajectories per task (400 total). We then applied SimpleVLA-RL directly to 3 completely different out-of-domain tasks (not exist in the training data and have no similar counterparts):
>
> | Task | Before RL | After RL |
> |------|-----------|----------|
> | Click Alarmclock | 69.5% | 81.3% |
> | Stack Bowls Two | 10.1% | 48.4% |
> | Grab Roller | 7.0% | 85.9% |
>
> As shown, despite no SFT on these 3 OOD tasks, π0.5 achieves meaningful zero-shot success rates, and direct RL yields substantial improvements without collecting any task-specific data. Note that the initial SFT on 8 RoboTwin tasks was necessary because π0.5 was originally trained on real-world data with ARX X5 embodiment, requiring SFT to adapt to simulation environment and Agilex embodiment used in RoboTwin. Although we used RoboTwin data, this experiment want to simulates a stronger VLA with zero-shot capabilities and provides preliminary validation of our hypothesis.
>
> **Future Outlook**: As VLA models continue to develop, we believe that in the near future (perhaps within a year), open-source VLA base models will be powerful enough to achieve reasonable zero-shot success rates on most common manipulation tasks. At that point, we can directly perform RL on base models without any data collection or warm-up procedures.
>
> ### References
>
> - [1] The Second Half
> - [2] $\pi_ {0.5}$: a Vision-Language-Action Model with Open-World Generalization

---

> > ### Comment · Reviewer_urcu · 2025-11-21
> >
> > Dear Authors,
> >
> > Thank you for your efforts to address weaknesses. All of my concerns have been addressed.
> >
> > In my opinion, In the field of VLA-RL, Using 8 A800 GPUs for 12-24 hours is a very cheap cost. I do think this work is excellent. I choose to update my score to 10. Hope your paper will be an oral paper.

---

> > > ### Author Response · Authors · 2025-11-21
> > >
> > > Dear Reviewer,
> > >
> > > Thank you very much for your recognition and your positive feedback! We will further refine and polish the manuscript according to your suggestions to make it as strong as possible.

---

> > ### Comment · Reviewer_urcu · 2025-11-21
> >
> > I also noticed that in recent works. And your explanation provide a brief summarization in VLA. It's good.

---

### Official Review · Reviewer_ogXC · 2025-10-30

**Soundness:** 3
**Presentation:** 3
**Contribution:** 3
**Rating:** 6
**Confidence:** 4

**Summary:**

This paper proposes **SimpleVLA-RL**, a framework designed to overcome the dependence of Vision-Language-Action (VLA) models on large-scale demonstration data during supervised fine-tuning. Building upon the veRL framework, the authors introduce tailored optimizations for reinforcement learning on autoregressive VLA models and validate their approach through both simulation benchmarks and real-world robotic experiments.

**Strengths:**

The paper is well-written, clearly structured, and presents a concise yet precise problem formulation. The experimental design is robust and the results are solid, demonstrating substantial improvements over existing state-of-the-art models across multiple benchmarks. Notably, the experiment using only a single demonstration during the RL phase provides compelling evidence that reinforcement learning can effectively mitigate the challenge of demonstration data scarcity.

**Weaknesses:**

Although the experimental results are comprehensive and detailed, the methodological innovation appears somewhat limited. The proposed parameter adjustments effectively improve performance, yet the paper does not sufficiently justify the suitability of directly transferring RL algorithms from large language models to VLA settings.

Moreover, the removal of the KL regularization term lacks supporting ablation studies, and the sparsity of the outcome-based reward may weaken the gradient signal, potentially affecting the stability of policy optimization. Furthermore, while the “Pushcut” phenomenon is intriguing and conceptually inspiring, it is demonstrated only in simulation, without quantitative validation in real-world settings—thus its generality and robustness remain uncertain.

**Questions:**

1. The current RL framework is validated primarily on autoregressive discrete-action VLAs. Have the authors considered extending SimpleVLA-RL to continuous-action VLA models?
2. Since the proposed method primarily relies on training in simulation and transferring to the real world, have the authors examined the potential sim-to-real gap?
3. The current generalization experiments focus primarily on task and scene transfer. Have the authors considered evaluating language generalization—for example, testing the model’s robustness to paraphrased or diverse natural-language commands within the same scene?
4. The authors remove the KL regularization term and adopt a sparse outcome-based reward. While this design may encourage greater exploration, it could also weaken policy constraints and optimization signals. Have the authors observed any instability or performance oscillation during training as a result?
5. The current RL training primarily focuses on in-domain tasks that share similar distributions with the supervised fine-tuning data. Have the authors considered performing RL training on out-of-domain tasks to further evaluate the model’s generalization ability and policy stability under distributional shifts?

---

> ### Author Response · Authors · 2025-11-21
>
> ## W1: Novelty
>
> **Simplicity as a feature, not a weakness.** We believe VLA RL training has been over-complicated. SimpleVLA-RL demonstrates that a surprisingly simple approach, sparse 0/1 rewards combined with GRPO, achieves SoTA results where more complex methods have struggled. This simplicity translates directly into practical advantages: no reward engineering, no critic networks, no reference model, and straightforward reproducibility. Reviewer urcu praised that our method *"removes reward-engineering and reference-model overhead"* and is *"explicit and easy to reproduce."* Reviewer EvC5 emphasized this is *"extremely simple and scalable, and I expect the community to adopt and build on the work"* and *"the first work to demonstrate reliable improvement with RL for VLAs in simulated domains."*
>
> **First to demonstrate reliable VLA-RL at scale.** Novelty is not solely about algorithmic complexity but about what works in practice. Despite the conceptual simplicity, no prior work has demonstrated our results: 99% success rate on LIBERO, 80% relative improvement on RoboTwin 2.0, strong data efficiency (single demonstration per task improves LIBERO-Long from 17.3% to 91.7%), improved generalization while SFT suffers catastrophic forgetting, and successful sim-to-real transfer (17.5%→38.5%). Reviewer EvC5 highlighted that we "*achieve substantial improvements*", "*generalize to out of distribution tasks*" and "*demonstrate these improvements actually transfer to the real world*". Importantly, even after DeepSeek-R1's release (February 2025), no other VLA-RL work has reproduced such consistent large-scale gains, largely due to the systems challenges we tackle.
>
> **Scalable framework with broad applicability.** Our contribution extends beyond the algorithm to provide an efficient, extensible VLA-RL infrastructure built on veRL: interactive closed-loop rollout with parallel multi-environment rendering, efficient trajectory sampling, and distributed training optimizations. The framework's versatility spans different benchmarks (LIBERO, RoboTwin), VLA architectures (autoregressive OpenVLA-OFT, diffusion-based π0.5 with 71.6%→91.3% improvement), and training scenarios (single-task, multi-task, out-of-domain tasks).
>
> **Practical impact and community validation.** Reviewer urcu (Confidence: 5) explicitly stated: *"In the field of VLA-RL, using 8 A800 GPUs for 12-24 hours is a very cheap cost. I do think this work is excellent. I choose to update my score to 10."* This reflects that the community values methods that are easy to put into practice. SimpleVLA-RL is novel in how it makes VLA-RL reliably work at scale through intentional simplicity, robust systems design, and comprehensive empirical validation.

---

> ### Author Response · Authors · 2025-11-21
>
> ## W2: The Suitability of Transferring RL Algorithms from LLMs to VLA
>
> Thank you for this valuable question. We think the suitability of transferring RL methods from LLMs to VLA settings from the following perspectives:
>
> **Model Architecture**:
> The core architecture of large language models is the Transformer. Current mainstream VLA models, whether OpenVLA-OFT [1], π0.5 [2], or the latest π0.6 [3], similarly use Transformer-based LLMs/VLMs as their backbone networks. For example, OpenVLA's core structure is based on LLaMA2, and π0.5's backbone is PaliGemma VLM. Therefore, LLMs and VLA models share fundamentally the same model architecture.
>
> **RL Methodology**:
> Although the two address different tasks, LLMs primarily handle mathematical reasoning and programming problems with text token outputs, while VLAs tackle manipulation tasks with action value outputs, their basic paradigm is entirely consistent: both involve sequentially generating a sequence given a problem. For instance, an LLM outputs a chain-of-thought reasoning sequence for a given problem, while a VLA outputs a multi-step action sequence for a manipulation task, with each text token or action value corresponding to a probability distribution.
>
> In summary, from the perspectives of model architecture and output format, RL for LLMs and RL for VLAs exhibit high consistency. More importantly, after transferring the RL method to VLA, we achieved significant performance improvements. We believe that substantial empirical gains is the strongest evidence of the method's suitability.
>
> ---
>
> ## W3: Ablation Studies on KL Regularization and the Stability of Training
>
> Thank you for pointing this out. We have added ablation experiments on KL regularization to Appendix D.1, including:  test tasks success rate vs. RL training steps curves before and after removing KL, as well as gradient norm, loss, and sampling success rate curves during training.
>
> **Experimental Results**:
> The experimental results demonstrate that removing KL regularization leads to a slight performance improvement, while training stability remains unaffected. The loss curves, gradient curves, and training sampling success rates all exhibit smooth progression and eventual convergence. Additionally, after removing KL regularization, the training framework becomes simpler since there is no need to compute reference model sampling probabilities and load an additional reference model, resulting in approximately 10% reduction in training time.
>
> **Additional Training Curves**:
> Furthermore, in Appendix D.2, we provide RL training curves for both OpenVLA-OFT and π0.5 on the RoboTwin 2.0 benchmark without KL regularization, including loss, gradient norm and sampling success rate. The results show no significant oscillations or instability throughout the training process.
>
> ---
>
> ## W4: Quantitative Validation of "Pushcut" in Real-World
>
> Thanks for this valuable suggestion! We did not validate the "Pushcut" phenomenon in real-world settings, which is precisely why we placed this experiment in the Discussion section of the paper.
>
> Validating "Pushcut" in the real world would require running online RL on a physical robot arm, which involves longer training times, more complex training framework, and higher costs, including:
>
> - Manual environment resets
> - Human-robot interaction during sampling
> - Continuous human supervision to prevent potential hardware damage
>
> These factors make real-robot RL experiments considerably more resource-intensive compared to simulation-based training.
>
> We will conduct more comprehensive real-robot RL experiments in future work on SimpleVLA-RL.
> We sincerely hope you understand these practical constraints.
>
> ---
>
> ## References
>
> [1] Fine-Tuning Vision-Language-Action Models: Optimizing Speed and Success
>
> [2] $\pi_ {0.5} $: a Vision-Language-Action Model with Open-World Generalization
>
> [3] $\pi_ {0.6} $: a VLA That Learns From Experience

---

> ### Author Response · Authors · 2025-11-21
>
> ## Q1: Extending SimpleVLA-RL to Continuous-Action VLAs
>
> In our paper, we use OpenVLA-OFT, an autoregressive model that generates discrete action tokens. To further demonstrate the generalizability of SimpleVLA-RL across different VLA architectures, we conducted additional experiments on Continuous-Action VLA. We do SimpleVLA-RL on π0.5 [1], a widely-used diffusion-based VLA using flow matching as action head to generate continuous action values.
> ### Method
>
> The algorithmic framework remains identical. The only modification is converting the flow matching process from an ODE (ordinary differential equation) to an SDE (stochastic differential equation) by injecting Gaussian noise. This modification is necessary because ODE-based generation is deterministic, the only stochasticity comes from the initial noise sample, whose probability does not contain model gradients and thus cannot be used for RL updates.
> We follow Flow-GRPO [2], which applies RL to flow matching in text-to-image generation, and adapt it to robotic manipulation tasks. Specifically, we convert the ODE process:
>
> $$
> \mathbf{x}\_{t+\Delta t} = \mathbf{x}\_t + \mathbf{v}\_\theta(\mathbf{x}\_t, t)\Delta t
> $$
>
> to an SDE process:
>
> $$
> \mathbf{x}\_{t+\Delta t} = \mathbf{x}\_t + \left[\mathbf{v}\_\theta(\mathbf{x}\_t, t) + \frac{\sigma\_t^2}{2t}\left(\mathbf{x}\_t + (1-t)\mathbf{v}\_\theta(\mathbf{x}\_t, t)\right)\right]\Delta t + \sigma\_t\sqrt{\Delta t}\epsilon
> $$
>
> where $\epsilon \sim \mathcal{N}(0, I)$ injects stochasticity. We use $\sigma\_t = a\sqrt{\frac{t}{1-t}}$, where $a$ is a scalar hyperparameter controlling the noise level.
>
> Under this SDE formulation, each transition follows a Gaussian distribution:
>
> $$
> \mathbf{x}\_{t+\Delta t} \sim \mathcal{N}\left(\boldsymbol{\mu}\_\theta(\mathbf{x}\_t, t), \sigma\_t^2 \Delta t \mathbf{I}\right)
> $$
>
> where the mean is:
>
> $$
> \boldsymbol{\mu}\_\theta(\mathbf{x}\_t, t) = \mathbf{x}\_t + \left[\mathbf{v}\_\theta(\mathbf{x}\_t, t) + \frac{\sigma\_t^2}{2t}\left(\mathbf{x}\_t + (1-t)\mathbf{v}\_\theta(\mathbf{x}\_t, t)\right)\right]\Delta t
> $$
>
> For a single transition $\mathbf{x}\_t \to \mathbf{x}\_{t+\Delta t}$, the log probability is:
>
> $$
> \log p(\mathbf{x}\_{t+\Delta t} \mid \mathbf{x}\_t) = -\log(\sigma\_t\sqrt{\Delta t}) - \frac{1}{2}\log(2\pi) - \frac{|\mathbf{x}\_{t+\Delta t} - \boldsymbol{\mu}\_\theta(\mathbf{x}\_t, t)|^2}{2\sigma\_t^2\Delta t}
> $$
>
> For the complete denoising trajectory $\mathbf{x}\_0 \to \mathbf{x}\_{\Delta t} \to \cdots \to \mathbf{x}\_1$, the log probability is:
>
> $$
> \log p\_\theta(\mathbf{x}\_1) = \log p\_0(\mathbf{x}\_0) + \sum\_{i=0}^{N-1} \log p(\mathbf{x}\_{t\_{i+1}} \mid \mathbf{x}\_{t\_i})
> $$
>
> This allows us to obtain the log probability of the action trajectory generated by policy $\pi\_\theta$.
>
> ### Results on RoboTwin 2.0
>
> | Model            | Lift Pot | Beat Hammer Block | Pick Dual Bottles | Place Phone Stand | Move Can Pot | Place A2B Left | Place Empty Cup | Handover Mic | Avg  |
> |------------------|----------|-------------------|-------------------|-------------------|--------------|----------------|-----------------|--------------|------|
> | RDT              | 45.0       | 22.0                | 18.0                | 13.0                | 33.0           | 21.0            | 42.0              | 95.0           | 36.1 |
> | π0.5             | 51.0       | 59.0                | 50.0                 | 22.0               | 41.0           | 38.0             | 84.0             | 96.0            | 55.1   |
> | OpenVLA-OFT      | 10.1     | 28.1              | 29.7              | 17.1              | 28.1         | 37.5           | 77.3            | 45.3         | 34.2 |
> | OpenVLA-OFT + RL | 64.1     | 87.5              | 68.3              | 39.6              | 61.2         | 45.3           | 94.2            | 89.2         | 68.7 |
> | π0.5             | 84.3     | 67.1              | 87.5              | 65.6              | 90.6         | 70.3           | 92.1            | 15           | 71.6 |
> | π0.5  + RL    | 96.8     | 88.5              | 95.3              | 82.8              | 95.3         | 75.2           | 100             | 96.8         | 91.3 |
> | Δ                | 12.5     | 21.4              | 7.8               | 17.2              | 4.7          | 4.9            | 7.9             | 81.8         | 19.7 |
>
> **Note:** OpenVLA-OFT uses single-task training with 1000 trajectories for SFT and 1000 scenarios for RL per task, whereas π0.5 uses multi-task training with only 50 trajectories per task for SFT and 50 scenarios per tasks for RL per task. Despite using only 5% data per task, π0.5 after SFT already surpasses OpenVLA-OFT after RL. With SimpleVLA-RL, π0.5 achieves further substantial improvements, reaching an average success rate of 91.3%.
>
> ### References
>
> [1] $\pi_ {0.5}$: a Vision-Language-Action Model with Open-World Generalization
>
> [2] Flow-GRPO: Training Flow Matching Models via Online RL

---

> ### Author Response · Authors · 2025-11-21
>
> ## Q2: Sim-to-Real Gap
>
> Thank you for this suggestion! Yes, we are fully aware of the sim-to-real gap. Whenever training is conducted in simulation, regardless of the simulation quality, transfer gaps are inevitable. For example, simulation environments like LIBERO lack sufficient physical realism and essentially cannot achieve sim-to-real transfer; while RoboTwin 2.0 [1], due to its excellent domain randomization design, has reasonable sim-to-real transfer capability, but gaps still exist.
>
> **Real-Robot Test Results**: In our real-robot tests on RoboTwin 2.0, we observed that the sim-to-real performance loss varies depending on the task and model used:
>
> | Model | Task | Sim Success Rate | Real Success Rate |
> |-------|------|------------------|-------------------|
> | OpenVLA-OFT | Stack Two Bowls | 76% | 70% |
> | OpenVLA-OFT | Click Bell | 90% | 60% |
> | OpenVLA-OFT | Place Bottle | 60% | 14% |
> | RDT | Place Bottle | 20% | 10% |
>
> Although the sim-to-real gap consistently exists, our real-robot experiments in Section 4.3 demonstrate that the performance improvements achieved through RL training in simulation successfully transfer to real-world deployment.
>
> **Regarding Method Generality**:
> It is important to emphasize that  SimpleVLA-RL is not limited to simulation environments. Our framework can inherently be used both in simulation and in real world. We chose to conduct experiments in simulation primarily for the following reasons:
>
> - Rapid validation of method effectiveness
> - Fair comparison with other methods/models on widely-used simulation benchmarks
> - Real-robot RL involves extensive hardware debugging and longer training cycles
>
> **Future Work**:
> We plan to extend SimpleVLA-RL to real-robot RL scenarios in future work. The current validation is primarily for efficiently demonstrating the method's feasibility, as the approach itself is not limited to simulation.
>
> ---
>
> ## Q3: Language Generalization
>
> Thank you for this valuable suggestion. Model robustness to language commands is indeed an important aspect of generalization. To verify SimpleVLA-RL's generalization to different prompts, we conducted the following experiments.
>
> **Experimental Setup**:
> We constructed seen and unseen prompts for two tasks in RoboTwin 2.0: "Lift Pot" and "Stack Two Bowls." Both seen and unseen prompts are task descriptions, but they differ in sentence structure and word choice.  For example, in "Stack Two Bowls":
>
> **Seen prompts include:**
> - "Lift {A}, put it on the table, and stack {B} above."
> - "Pick up {A}, place it down, then position {B} on top."
> - "Set {B} over {A} to create the stack."
> - "Stack {B} securely on top of {A}."
>
> **Unseen prompts include:**
> - "Take {A}, set it down, and stack {B} on top."
> - "Place {B} carefully over {A} to stack them."
> - "Stack {B} directly over {A}."
>
> **Training Procedure**:
> We SFT OpenVLA-OFT on seen prompts, then performed RL on the SFT model using seen prompts. All other training settings are consistent with Table2 in our paper. RL training continued until convergence, and we evaluated the RL-finetuned OpenVLA-OFT using both seen and unseen prompts in the same test scenarios.
>
> **Results**:
>
> | Task | Model | Seen Prompt | Unseen Prompt |
> |------|-------|-------------|---------------|
> | Lift Pot | SFT OpenVLA-OFT | 10% | 8% |
> | Lift Pot | RL OpenVLA-OFT | 66% | 62% |
> | Stack Two Bowls | SFT OpenVLA-OFT | 41% | 38% |
> | Stack Two Bowls | RL OpenVLA-OFT | 73% | 66% |
>
> **Conclusion**:
> The results demonstrate that after RL training, the model maintains strong robustness to varied language instructions, with only minor performance differences between seen and unseen prompts.
>
> ---
>
> ## Q4:  instability & performance oscillation
>
> Thank you for pointing this out. We have added ablation experiments on KL regularization to Appendix D.1, including: test tasks success rate vs. RL training steps curves before and after removing KL, as well as gradient norm, loss, and sampling success rate curves during training.
>
> The experimental results demonstrate that removing KL regularization leads to a slight performance improvement, while training stability remains unaffected. The loss curves, gradient curves, and training sampling success rates all exhibit smooth progression and eventual convergence. Additionally, after removing KL regularization, the training framework becomes simpler since there is no need to compute reference model sampling probabilities and load an additional reference model, resulting in approximately 10% reduction in training time.
>
> Furthermore, in Appendix D.2, we provide RL training curves for both OpenVLA-OFT and π0.5 on the RoboTwin 2.0 benchmark without KL regularization, including loss, gradient norm and sampling success rate. The results show no significant oscillations or instability throughout the training process.
>
> ### References
>
> [1] RoboTwin 2.0: A Scalable Data Generator and Benchmark with Strong Domain Randomization for Robust Bimanual Robotic Manipulation

---

> ### Author Response · Authors · 2025-11-21
>
> ## Q5: Performing RL on Out-of-Domain Tasks
>
> Thank you for this excellent suggestion. Currently, all our experiments perform RL on in-domain tasks where the RL tasks share the same distribution as the SFT data. Investigating RL on out-of-domain tasks that differ from the training data distribution would more comprehensively evaluate the generalization and applicability of SimpleVLA-RL.
> Therefore, we conducted the following experiments.
>
> **Experimental Setup**:
> We first performed multi-tasks SFT on the π0.5 base model using 8 tasks from RoboTwin 2.0 [1]: *Lift Pot*, *Beat Hammer Block*, *Pick Dual Bottles*, *Place Phone Stand*, *Move Can Pot*, *Place A2B Left*, *Place Empty Cup*, *and* *Handover Mic*. Specifically, we collected 50 trajectories per task, totaling 400 trajectories for multi-tasks SFT. (Detailed task descriptions and demonstrations can be found at official doc of RoboTwin 2.0)
>
> We then used the finetuned π0.5 to perform RL on 3 completely different out-of-domain tasks: *Click Alarmclock*, *Grab Roller*, *and* *Stack Bowls Two*. For each task, RL training was conducted independently using 50 simulation scenarios.
>
> **Results**:
>
> | Task | Before RL | After RL |
> |------|-----------|----------|
> | Click Alarmclock | 69.5% | 81.3% |
> | Stack Bowls Two | 10.1% | 48.4% |
> | Grab Roller | 7.0% | 85.9% |
>
> **Conclusion**:
> The results demonstrate that SimpleVLA-RL achieves substantial performance improvements even when performing RL on out-of-domain tasks that differ from the SFT data distribution, enhancing model generalization under distributional shifts. Furthermore, the training curves for OOD tasks exhibit similar trends to those of in-distribution tasks, with stable training and normal convergence.
>
> ### References
>
> [1] RoboTwin 2.0: A Scalable Data Generator and Benchmark with Strong Domain Randomization for Robust Bimanual Robotic Manipulation

---

### Official Review · Reviewer_EvC5 · 2025-10-31

**Soundness:** 3
**Presentation:** 4
**Contribution:** 3
**Rating:** 8
**Confidence:** 4

**Summary:**

The paper demonstrates how advances in LLM post-training can be ported over to robotics domains by finetuning VLA models in simulation with simple modifications to GRPO. Specifically, the paper introduces best practices to encourage exploration and ensure stable training, such as dynamic sampling of batches to ensure only informative gradients are propagated, best practices for clipping policy updates, and high-temperature sampling. The approach only uses sparse rewards. The paper demonstrates impressive improvements over SFT, and then demonstrates these improvements hold when transferring back to the real world.

**Strengths:**

$\textbf{Simplicity + Scalability}$: The approach is extremely simple and scalable, and I expect the community to adopt and build on the work. Specifically, the approach is much simpler than recently proposed techniques for training diffusion-based policy heads, and opens the door for porting over new insights from LLM finetuning to robotics.

$\textbf{Strong Results}$:  To my knowledge, this is the first work to demonstrate reliable improvement with RL for VLAs in simulated domains. The paper demonstrates the ability of the method to achieve substantial improvements in success rates across a range of tasks, demonstrates some ability to generalize to out of distribution tasks, and demonstrates these improvements actually transfer to the real world.

$\textbf{Thorough Evaluation}$: The paper has a strong evaluation and clearly conveys the importance of each of the design decisions.

$\textbf{Timliness:}$ In short, this is a solid, timley submission and an easy acceptance for me.

**Weaknesses:**

$\textbf{Additional Related Work}$: The discussion of related work focuses primarily on other approaches for post-training VLAs. However, the paper would be strengthened by additional discussion one recent works which finetune diffusion policy heads, such as Diffusion Policy Policy Optimization. We know that modifications to GRPO are effective at post-training LLMs, but to me the most surprising thing about this paper is that it was able to stably optimize through a diffusion head. Some additional intuition on why this simple approach works so well would be nice.

**Questions:**

Can the authors provide some intuition for why they were able to effectively optimize through diffusion-based policies with such a simple approach?

---

> ### Author Response · Authors · 2025-11-21
>
> ## W1: Additional Related Work on Diffusion Policy RL
>
> Thank you for this valuable suggestion. Our related work section lacks discussion of RL methods for diffusion policy. We will add references to relevant works such as DPPO [1], Flow-GRPO [2] in the revised manuscript.
>
> ---
>
> ## Clarification on Our Experimental Setup
>
> We would like to clarify that in our experiments, OpenVLA-OFT uses the LLaMA2 output head, which outputs logits for action tokens and then samples specific action tokens via temperature sampling to obtain action values. We did not use OFT's default MLP action head because it directly outputs deterministic actions without any stochasticity, making it impossible to compute the probability of generated action, and thus incompatible with PPO-like reinforcement learning. We replaced it with the LLaMA2 output head to directly generate logits and obtain action probabilities. This is mentioned in Section 2.1 (Interactive VLA Rollout) and Appendix F.2 (Backbone Modification Details).
>
> ---
>
> ## Extending SimpleVLA-RL to Diffusion-based VLAs
>
> This raises a natural question: OpenVLA-OFT is an autoregressive model that generates discrete action tokens, but other VLA models such as π0 and $\pi_ {0.5}$ [3] use diffusion-based action heads (e.g., flow matching) to generate continuous action values. Can SimpleVLA-RL be applied to these VLAs?
>
> **The answer is yes.** We have successfully applied SimpleVLA-RL to π0.5, a widely-used and high-performing diffusion-based VLA.  In the following comment, we present our detailed method and experimental results.
>
> ### References
>
> [1] Diffusion Policy Policy Optimization
>
> [2] Flow-GRPO: Training Flow Matching Models via Online RL
>
> [3] $\pi_ {0.5}$: a Vision-Language-Action Model with Open-World Generalization

---

> ### Author Response · Authors · 2025-11-21
>
> ## Extending SimpleVLA-RL to Diffusion-based VLAs
> We do SimpleVLA-RL on π0.5 [1], a widely-used and high-performing diffusion-based VLA model using flow matching as action head to generate continuous action values.
> ### Method
>
> The algorithmic framework remains identical. The only modification is converting the flow matching process from an ODE (ordinary differential equation) to an SDE (stochastic differential equation) by injecting Gaussian noise. This modification is necessary because ODE-based generation is deterministic, the only stochasticity comes from the initial noise sample, whose probability does not contain model gradients and thus cannot be used for RL updates.
> We follow Flow-GRPO [2], which applies RL to flow matching in text-to-image generation, and adapt it to robotic manipulation tasks. Specifically, we convert the ODE process:
>
> $$
> \mathbf{x}\_{t+\Delta t} = \mathbf{x}\_t + \mathbf{v}\_\theta(\mathbf{x}\_t, t)\Delta t
> $$
>
> to an SDE process:
>
> $$
> \mathbf{x}\_{t+\Delta t} = \mathbf{x}\_t + \left[\mathbf{v}\_\theta(\mathbf{x}\_t, t) + \frac{\sigma\_t^2}{2t}\left(\mathbf{x}\_t + (1-t)\mathbf{v}\_\theta(\mathbf{x}\_t, t)\right)\right]\Delta t + \sigma\_t\sqrt{\Delta t}\epsilon
> $$
>
> where $\epsilon \sim \mathcal{N}(0, I)$ injects stochasticity. We use $\sigma\_t = a\sqrt{\frac{t}{1-t}}$, where $a$ is a scalar hyperparameter controlling the noise level.
>
> Under this SDE formulation, each transition follows a Gaussian distribution:
>
> $$
> \mathbf{x}\_{t+\Delta t} \sim \mathcal{N}\left(\boldsymbol{\mu}\_\theta(\mathbf{x}\_t, t), \sigma\_t^2 \Delta t \mathbf{I}\right)
> $$
>
> where the mean is:
>
> $$
> \boldsymbol{\mu}\_\theta(\mathbf{x}\_t, t) = \mathbf{x}\_t + \left[\mathbf{v}\_\theta(\mathbf{x}\_t, t) + \frac{\sigma\_t^2}{2t}\left(\mathbf{x}\_t + (1-t)\mathbf{v}\_\theta(\mathbf{x}\_t, t)\right)\right]\Delta t
> $$
>
> For a single transition $\mathbf{x}\_t \to \mathbf{x}\_{t+\Delta t}$, the log probability is:
>
> $$
> \log p(\mathbf{x}\_{t+\Delta t} \mid \mathbf{x}\_t) = -\log(\sigma\_t\sqrt{\Delta t}) - \frac{1}{2}\log(2\pi) - \frac{|\mathbf{x}\_{t+\Delta t} - \boldsymbol{\mu}\_\theta(\mathbf{x}\_t, t)|^2}{2\sigma\_t^2\Delta t}
> $$
>
> For the complete denoising trajectory $\mathbf{x}\_0 \to \mathbf{x}\_{\Delta t} \to \cdots \to \mathbf{x}\_1$, the log probability is:
>
> $$
> \log p\_\theta(\mathbf{x}\_1) = \log p\_0(\mathbf{x}\_0) + \sum\_{i=0}^{N-1} \log p(\mathbf{x}\_{t\_{i+1}} \mid \mathbf{x}\_{t\_i})
> $$
>
> This allows us to obtain the log probability of the action trajectory generated by policy $\pi\_\theta$.
>
> ### Results on RoboTwin 2.0
>
> | Model            | Lift Pot | Beat Hammer Block | Pick Dual Bottles | Place Phone Stand | Move Can Pot | Place A2B Left | Place Empty Cup | Handover Mic | Avg  |
> |------------------|----------|-------------------|-------------------|-------------------|--------------|----------------|-----------------|--------------|------|
> | RDT              | 45.0       | 22.0                | 18.0                | 13.0                | 33.0           | 21.0            | 42.0              | 95.0           | 36.1 |
> | π0.5             | 51.0       | 59.0                | 50.0                 | 22.0               | 41.0           | 38.0             | 84.0             | 96.0            | 55.1   |
> | OpenVLA-OFT      | 10.1     | 28.1              | 29.7              | 17.1              | 28.1         | 37.5           | 77.3            | 45.3         | 34.2 |
> | OpenVLA-OFT + RL | 64.1     | 87.5              | 68.3              | 39.6              | 61.2         | 45.3           | 94.2            | 89.2         | 68.7 |
> | π0.5             | 84.3     | 67.1              | 87.5              | 65.6              | 90.6         | 70.3           | 92.1            | 15           | 71.6 |
> | π0.5  + RL    | 96.8     | 88.5              | 95.3              | 82.8              | 95.3         | 75.2           | 100             | 96.8         | 91.3 |
> | Δ                | 12.5     | 21.4              | 7.8               | 17.2              | 4.7          | 4.9            | 7.9             | 81.8         | 19.7 |
>
> **Note:** OpenVLA-OFT uses single-task training with 1000 trajectories for SFT and 1000 scenarios for RL per task, whereas π0.5 uses multi-task training with only 50 trajectories per task for SFT and 50 scenarios per tasks for RL per task. Despite using only 5% data per task, π0.5 after SFT already surpasses OpenVLA-OFT after RL. With SimpleVLA-RL, π0.5 achieves further substantial improvements, reaching an average success rate of 91.3%.
>
> ### References
>
> [1] $\pi_ {0.5}$: a Vision-Language-Action Model with Open-World Generalization
>
> [2] Flow-GRPO: Training Flow Matching Models via Online RL

---

> ### Author Response · Authors · 2025-11-21
>
> ## Question: Why SimpleVLA-RL Works for Diffusion-based VLAs
>
> The reviewer raises an excellent point. The key intuition lies in how PPO-like algorithms update models through their objective function (e.g., GRPO):
>
> $$
> J\_{\text{GRPO}}(\theta) = \mathbb{E}\_{s\_0 \sim \mathcal{D}, \{\tau\_i\} \sim \pi\_{\theta\_{\text{old}}}} \left[ \frac{1}{G} \sum\_{i=1}^G \frac{1}{|\tau\_i|} \sum\_{t=1}^{|\tau\_i|} \min\left( r\_{i,t}(\theta) \hat{A}\_i, \text{clip}(r\_{i,t}(\theta), 1-\epsilon, 1+\epsilon) \hat{A}\_i \right) - \beta D\_{\text{KL}}(\pi\_\theta \| \pi\_{\text{ref}}) \right]
> $$
>
> where $r\_{i,t}(\theta) = \frac{\pi\_\theta(a\_{i,t}|s\_{i,t})}{\pi\_{\theta\_{\text{old}}}(a\_{i,t}|s\_{i,t})}$ is the probability ratio.
>
> During optimization:
>
> * **Positive rewards:** Actions from successful trajectories receive positive advantages $\hat{A}\_i > 0$, increasing their log probability $\log p(a)$ through the ratio $r\_{i,t}(\theta)$
> * **Negative rewards:** Actions from failed trajectories receive negative advantages $\hat{A}\_i < 0$, decreasing their log probability
>
> This mechanism naturally optimizes the model distribution by increasing the probability of correct actions and decreasing incorrect actions.
>
> **For Autoregressive VLAs:** The action probability directly corresponds to token generation probabilities, so the model learns to generate correct action tokens more frequently.
>
> **For Diffusion-based VLAs (e.g., π0.5):** The log probability is:
>
> $$
> \log p(\mathbf{x}\_{t+\Delta t} \mid \mathbf{x}\_t) = -\log(\sigma\_t\sqrt{\Delta t}) - \frac{1}{2}\log(2\pi) - \frac{|\mathbf{x}\_{t+\Delta t} - \boldsymbol{\mu}\_\theta(\mathbf{x}\_t, t)|^2}{2\sigma\_t^2\Delta t}
> $$
>
> Only the $-|\mathbf{x}\_{t+\Delta t} - \boldsymbol{\mu}\_\theta(\mathbf{x}\_t, t)|^2$ has gradients with policy $\theta$.
>
> Therefore:
>
> * **For correct actions (positive reward):** Maximizing $\log p$ minimizes $|\mathbf{x}\_{t+\Delta t} - \boldsymbol{\mu}\_\theta|^2$, pulling the predicted mean $\boldsymbol{\mu}\_\theta$ closer to the rewarded action $\mathbf{x}\_{t+\Delta t}$
> * **For incorrect actions (negative reward):** Minimizing $\log p$ increases $|\mathbf{x}\_{t+\Delta t} - \boldsymbol{\mu}\_\theta|^2$, pushing the model to generate actions different from the penalized ones
>
> This simple gradient-based mechanism enables the diffusion model to learn the correct action distribution, analogous to how autoregressive models adjust token probabilities.
>
> Similar insights on why GRPO effectively optimizes flow matching models can also be found in the Flow-GRPO paper.

---

### Official Review · Reviewer_qg2P · 2025-11-01

**Soundness:** 3
**Presentation:** 3
**Contribution:** 3
**Rating:** 6
**Confidence:** 4

**Summary:**

This paper introduces SimpleVLA-RL, an RL framework built on top of VeRL to train Vision-Language-Action (VLA) models for robotic manipulation tasks. The work addresses two fundamental challenges in current VLA training: (1) the scarcity and high cost of large-scale robotic trajectory data required for supervised fine-tuning (SFT), and (2) limited generalization to tasks under distribution shift. SimpleVLA-RL leverages all recent findings from LLM RL finetuning literature such as GRPO with dynamic sampling, variable clipping threshold, high temperature rollout sampling and no KL penalty to build a stable RL finetuning setup for VLA training. The paper itself does not introduce any new ideas or algorithms to enable RL training for VLA but shows that findings from LLM community generalize to robotics without any modifications.

**Strengths:**

1. Application of RL for training low-level control policies for manipulation to scale VLA training without relying on  large amounts of demonstration data.
2. The experimental results clearly show performance improvements achievable through RL training are significant
3. The analysis section is interesting and studies valuable questions with well thought out experiments

**Weaknesses:**

1. The experiments section doesn’t mention details about the training datasets and benchmarks. From the current description it is a bit unclear to me on what task, episodes, and bechmarks was the RL finetuning for the VLA policies done? My understanding is OpenVLA-OFT was first trained on real world robotics datasets and then for benchmarking on libero it was finetuned with small amounts of simulation only trajectories. I would like to understand for experiments presented in tables 1, 2 and 3 did the authors use the SFT finetuned OpenVLA-OFT model as initial checkpoint for RL or RL is performed directly on base model (aka not finetuned on sim data with SFT).
2. Another concern I have is the SFT finetuned results for full libero dataset experiment used by OpenVLA-OFT in the paper does not match the results presented in the original paper table 2 here https://arxiv.org/pdf/2502.19645. I would like to ask authors to clarify why there is such a big gap (ex: Spatial SR is 96.2 in original paper vs 91.6 in table 3 and 4).
3. The experiments presented in analysis section 4.2 are definitely interesting and valuable. However, the details about how the amount of compute/data used for both SFT and RL is a confounding variable that is unaccounted for. Can authors please describe the exact amount of model updates, trajectories used for this comparison? I believe such a comparison only makes sense when done under a fixed budget of training updates to get a clear idea.

**Questions:**

On L177 authors claim that critic free RL algorithms suffer from vanishing gradients problem. Can authors add a supporting citation for this? I am unsure how true this claim is. Critic free RL algorithms suffer from high-variance but there is no inherent issue in these algorithms that causes vanishing gradients. If there are no supporting citations for this statement I would suggest the authors to remove this line.

This paper does not propose any novel ideas to enable RL for VLAs. It can be seen as more of application of recent techniques used in RL finetuning for LLMs applied directly to VLA finetuning. However, it does have valuable experiments and insights that most of the techniques used in making RL training stable and effective for LLMs generalize to VLA finetuning in the current setup. That is why I recommend the paper to be accepted conditioned that authors address the above mentioned weaknesses and minor issues in some claims made.

---

> ### Author Response · Authors · 2025-11-21
>
> ## W1: Clarification on experimental setup and initialization
>
> Thank you for this question. Your understanding is correct. In Tables 1, 2, and 3, the initial checkpoint for RL is the SFT-finetuned OpenVLA-OFT (finetuned on simulation data). We provide detailed experimental setups below:
>
> ### LIBERO experiments (Table 3)
>
> For each task suite (e.g., LIBERO-Long), we first SFT-finetune the OpenVLA-OFT base model using 500 simulation trajectories provided by LIBERO, yielding four separate SFT-finetuned checkpoints. These checkpoints are evaluated on the test scenarios of their corresponding task suites, achieving success rates of 91.6, 95.3, 90.6, and 86.5 respectively (as shown in Table 3). We then perform RL on each SFT-finetuned checkpoint using 500 simulation scenarios per task suite, resulting in the RL-finetuned models with success rates of 99.4, 99.1, 99.2, and 98.5 respectively.
>
> ### RoboTwin 1.0 & 2.0 experiments (Tables 1 & 2)
>
> For each individual task, we first collect simulation trajectories to SFT-finetune the OpenVLA-OFT base model, 50 trajectories per task for RoboTwin 1.0, and 1000 trajectories per task for RoboTwin 2.0 (due to increased scene complexity from randomization). Each task's SFT-finetuned checkpoint is evaluated on 100 test scenarios. Subsequently, we perform RL on each SFT-finetuned checkpoint using task-specific simulation scenarios, 100 scenarios per task for RoboTwin 1.0 and 1000 scenarios for RoboTwin 2.0, and evaluate the RL-finetuned models on the same test scenarios.
>
> We will add comprehensive details about training datasets, benchmarks, and initialization procedures in the revised manuscript.
>
> ---
>
> ## W2: Discrepancy in OpenVLA-OFT baseline results
>
> Thank you for pointing this out. The discrepancy arises because we made simplifications to the model inputs and architecture compared to the official OpenVLA-OFT implementation. These modifications are detailed in Appendix F.2 (Backbone Modification Details). We explain the reason and necessity below.
>
> ### Model inputs
>
> We utilize only single-view images, language instructions, and robot proprioceptive states as model inputs, whereas the official model additionally incorporates wrist camera images. Furthermore, for LIBERO experiments, we do not include robot proprioceptive states in model inputs. These simplifications were made to achieve improved training and inference efficiency.
>
> ### Model architecture
>
> We employ only the parallel decoding and action chunking designs (which the OFT paper's experiments validated as the primary contributors to performance gains). Additionally, we use the LLaMA2 output head to generate action tokens with cross-entropy loss, whereas the official model uses an MLP to generate continuous actions with L1 regression loss. This modification was necessary because reinforcement learning requires computing action probabilities, whereas an MLP action head can only output deterministic continuous action values without being able to compute the probability.
>
> Therefore, our experimental setting corresponds to the configuration in the OFT paper Table 1 with "Policy inputs: third-person image, language instruction" and "OpenVLA (fine-tuned) + PD&AC" [1], which reports results of 91.3, 92.7, 90.5, and 86.5 (official), nearly identical to our reported results 91.6, 95.3, 90.6, and 86.5.
>
> We will clarify this more explicitly in the revised vision to avoid confusion.
>
> ---
>
> ## W3: Compute and data budget for SFT vs RL comparison
>
> Thank you for this valuable suggestion. Clarifying the compute and data budget is indeed important for a fair comparison between SFT and RL.
>
> In the generalization analysis experiments (Section 4.2), both SFT and RL use the OpenVLA-OFT model finetuned under the "One-Trajectory SFT" setting as the initial checkpoint. For each experimental group, 9 tasks are randomly selected as training tasks (seen tasks), and the remaining task serves as the evaluation task (unseen task).
>
> ### Data and compute details
>
> |   | SFT | RL |
> |---|--|---|
> | **Training Data**  | 450 demonstration trajectories (50 per task)  | 0 demonstration trajectories  and 450 simulation scenarios (50 per task)   |
> | **Training Steps**  | ~70k steps   | ~60 steps  |
> | **Model Update Time**   | ~20 hours   | ~8 hours   |
> | **Trajectory Sampling Time** | N/A   | ~22 hours (limited by simulator rendering speed)  |
> | **Total Training Time**  | ~20 hours   | ~30 hours  |
> | **Hardware**  | 8× A800-80GB | 8× A800-80GB    |
>
> Both SFT and RL are trained until the model performance on training tasks is fully converged.
>
> We will include detailed specifications of model updates, trajectories, and compute budgets in the revised vision.
>
> ---
>
>
> ### References
>
> [1] Fine-Tuning Vision-Language-Action Models: Optimizing Speed and Success

---

> ### Author Response · Authors · 2025-11-21
>
> ## Q1: Citation for vanishing gradients in critic-free RL
>
> We sincerely thank the reviewer for highlighting this issue, which arose mainly from a lack of detailed explanation that could cause confusion. You are correct that critic-free RL algorithms do not inherently suffer from vanishing gradients. However, a specific issue can arise when using sparse 0/1 outcome rewards.
> For example, in GRPO, if all sampled outputs $\\{o_i\\}_{i=1}^G$ for a particular task are incorrect and receive the same reward of 0, the resulting advantage for this group becomes zero, leading to no gradients for policy updates. This issue is also discussed in Section 3.2 of DAPO [1].
>
> This issue occurs very frequently in our experiments because the backbone we use, OpenVLA-OFT, has weak capabilities and priors. As a result, it often fails to rollout correct trajectories, causing all sampled trajectories within a group to be incorrect and ultimately leading to the problem described above.
>
> ---
>
> ## Q2: Novelty
>
> **Simplicity as a feature, not a weakness.** We believe VLA RL training has been over-complicated. SimpleVLA-RL demonstrates that a surprisingly simple approach, sparse 0/1 rewards combined with GRPO, achieves SoTA results where more complex methods have struggled. This simplicity translates directly into practical advantages: no reward engineering, no critic networks, no reference model, and straightforward reproducibility. Reviewer urcu praised that our method *"removes reward-engineering and reference-model overhead"* and is *"explicit and easy to reproduce."* Reviewer EvC5 emphasized this is *"extremely simple and scalable, and I expect the community to adopt and build on the work"* and *"the first work to demonstrate reliable improvement with RL for VLAs in simulated domains."*
>
> **First to demonstrate reliable VLA-RL at scale.** Novelty is not solely about algorithmic complexity but about what works in practice. Despite the conceptual simplicity, no prior work has demonstrated our results: 99% success rate on LIBERO, 80% relative improvement on RoboTwin 2.0, strong data efficiency (single demonstration per task improves LIBERO-Long from 17.3% to 91.7%), improved generalization while SFT suffers catastrophic forgetting, and successful sim-to-real transfer (17.5%→38.5%). Reviewer EvC5 highlighted that we "*achieve substantial improvements*", "*generalize to out of distribution tasks*" and "*demonstrate these improvements actually transfer to the real world*". Importantly, even after DeepSeek-R1[2]'s release (February 2025), no other VLA-RL work has reproduced such consistent large-scale gains, largely due to the systems challenges we tackle.
>
> **Scalable framework with broad applicability.** Our contribution extends beyond the algorithm to provide an efficient, extensible VLA-RL infrastructure built on veRL[3]: interactive closed-loop rollout with parallel multi-environment rendering, efficient trajectory sampling, and distributed training optimizations. The framework's versatility spans different benchmarks (LIBERO, RoboTwin), VLA architectures (autoregressive OpenVLA-OFT, diffusion-based π0.5 with 71.6%→91.3% improvement), and training scenarios (single-task, multi-task, out-of-domain tasks).
>
> **Practical impact and community validation.** Reviewer urcu (Confidence: 5) explicitly stated: *"In the field of VLA-RL, using 8 A800 GPUs for 12-24 hours is a very cheap cost. I do think this work is excellent. I choose to update my score to 10."* This reflects that the community values methods that are easy to put into practice. SimpleVLA-RL is novel in how it makes VLA-RL reliably work at scale through intentional simplicity, robust systems design, and comprehensive empirical validation.
>
> ---
>
> ### References
>
> [1] DAPO: An Open-Source LLM Reinforcement Learning System at Scale
>
> [2] DeepSeek-R1: Incentivizing Reasoning Capability in LLMs via Reinforcement Learning
>
> [3] verl: Volcano Engine Reinforcement Learning for LLMs

---

> ### Author Response · Authors · 2025-11-21
>
> ## Additional Experiments: Extending SimpleVLA-RL to Diffusion-based VLAs
>
> In our paper, we use OpenVLA-OFT, an autoregressive model that generates discrete action tokens. To further demonstrate the generalizability of SimpleVLA-RL across different VLA architectures, we conducted additional experiments on Diffusion-based VLA. We do SimpleVLA-RL on π0.5 [1], a widely-used diffusion-based VLA using flow matching as action head to generate continuous action values.
>
> ### Method
>
> The algorithmic framework remains identical. The only modification is converting the flow matching process from an ODE (ordinary differential equation) to an SDE (stochastic differential equation) by injecting Gaussian noise. This modification is necessary because ODE-based generation is deterministic, the only stochasticity comes from the initial noise sample, whose probability does not contain model gradients and thus cannot be used for RL updates.
> We follow Flow-GRPO [2], which applies RL to flow matching in text-to-image generation, and adapt it to robotic manipulation tasks. Specifically, we convert the ODE process:
>
> $$
> \mathbf{x}\_{t+\Delta t} = \mathbf{x}\_t + \mathbf{v}\_\theta(\mathbf{x}\_t, t)\Delta t
> $$
>
> to an SDE process:
>
> $$
> \mathbf{x}\_{t+\Delta t} = \mathbf{x}\_t + \left[\mathbf{v}\_\theta(\mathbf{x}\_t, t) + \frac{\sigma\_t^2}{2t}\left(\mathbf{x}\_t + (1-t)\mathbf{v}\_\theta(\mathbf{x}\_t, t)\right)\right]\Delta t + \sigma\_t\sqrt{\Delta t}\epsilon
> $$
>
> where $\epsilon \sim \mathcal{N}(0, I)$ injects stochasticity. We use $\sigma\_t = a\sqrt{\frac{t}{1-t}}$, where $a$ is a scalar hyperparameter controlling the noise level.
>
> Under this SDE formulation, each transition follows a Gaussian distribution:
>
> $$
> \mathbf{x}\_{t+\Delta t} \sim \mathcal{N}\left(\boldsymbol{\mu}\_\theta(\mathbf{x}\_t, t), \sigma\_t^2 \Delta t \mathbf{I}\right)
> $$
>
> where the mean is:
>
> $$
> \boldsymbol{\mu}\_\theta(\mathbf{x}\_t, t) = \mathbf{x}\_t + \left[\mathbf{v}\_\theta(\mathbf{x}\_t, t) + \frac{\sigma\_t^2}{2t}\left(\mathbf{x}\_t + (1-t)\mathbf{v}\_\theta(\mathbf{x}\_t, t)\right)\right]\Delta t
> $$
>
> For a single transition $\mathbf{x}\_t \to \mathbf{x}\_{t+\Delta t}$, the log probability is:
>
> $$
> \log p(\mathbf{x}\_{t+\Delta t} \mid \mathbf{x}\_t) = -\log(\sigma\_t\sqrt{\Delta t}) - \frac{1}{2}\log(2\pi) - \frac{|\mathbf{x}\_{t+\Delta t} - \boldsymbol{\mu}\_\theta(\mathbf{x}\_t, t)|^2}{2\sigma\_t^2\Delta t}
> $$
>
> For the complete denoising trajectory $\mathbf{x}\_0 \to \mathbf{x}\_{\Delta t} \to \cdots \to \mathbf{x}\_1$, the log probability is:
>
> $$
> \log p\_\theta(\mathbf{x}\_1) = \log p\_0(\mathbf{x}\_0) + \sum\_{i=0}^{N-1} \log p(\mathbf{x}\_{t\_{i+1}} \mid \mathbf{x}\_{t\_i})
> $$
>
> This allows us to obtain the log probability of the action trajectory generated by policy $\pi\_\theta$.
>
> ### Results on RoboTwin 2.0
>
> | Model            | Lift Pot | Beat Hammer Block | Pick Dual Bottles | Place Phone Stand | Move Can Pot | Place A2B Left | Place Empty Cup | Handover Mic | Avg  |
> |------------------|----------|-------------------|-------------------|-------------------|--------------|----------------|-----------------|--------------|------|
> | RDT              | 45.0       | 22.0                | 18.0                | 13.0                | 33.0           | 21.0            | 42.0              | 95.0           | 36.1 |
> | π0.5             | 51.0       | 59.0                | 50.0                 | 22.0               | 41.0           | 38.0             | 84.0             | 96.0            | 55.1   |
> | OpenVLA-OFT      | 10.1     | 28.1              | 29.7              | 17.1              | 28.1         | 37.5           | 77.3            | 45.3         | 34.2 |
> | OpenVLA-OFT + RL | 64.1     | 87.5              | 68.3              | 39.6              | 61.2         | 45.3           | 94.2            | 89.2         | 68.7 |
> | π0.5             | 84.3     | 67.1              | 87.5              | 65.6              | 90.6         | 70.3           | 92.1            | 15           | 71.6 |
> | π0.5  + RL    | 96.8     | 88.5              | 95.3              | 82.8              | 95.3         | 75.2           | 100             | 96.8         | 91.3 |
> | Δ                | 12.5     | 21.4              | 7.8               | 17.2              | 4.7          | 4.9            | 7.9             | 81.8         | 19.7 |
>
> **Note:** OpenVLA-OFT uses single-task training with 1000 trajectories for SFT and 1000 scenarios for RL per task, whereas π0.5 uses multi-task training with only 50 trajectories per task for SFT and 50 scenarios per tasks for RL per task. Despite using only 5% data per task, π0.5 after SFT already surpasses OpenVLA-OFT after RL. With SimpleVLA-RL, π0.5 achieves further substantial improvements, reaching an average success rate of 91.3%.
>
> ### References
>
> [1] $\pi_ {0.5}$: a Vision-Language-Action Model with Open-World Generalization
>
> [2] Flow-GRPO: Training Flow Matching Models via Online RL

---

> > ### Comment · Reviewer_qg2P · 2025-11-25
> > **Response to authors**
> >
> > Thanks for the detailed response and additional results. Overall I think this is a good paper with interesting experiments. The response from authors have addressed my concerns so I updated my rating. I recommend authors to add all the missing details in text and additional results discussed to the paper. The paper has interesting contributions but writing is missing a bunch of the important details which are required for readers to fully understand experimental setup and baselines.
> >
> > > We sincerely thank the reviewer for highlighting this issue, which arose mainly from a lack of detailed explanation that could cause confusion. You are correct that critic-free RL algorithms do not inherently suffer from vanishing gradients. However, a specific issue can arise when using sparse 0/1 outcome rewards. For example, in GRPO, if all sampled outputs
> >  for a particular task are incorrect and receive the same reward of 0, the resulting advantage for this group becomes zero, leading to no gradients for policy updates. This issue is also discussed in Section 3.2 of DAPO [1].
> > > This issue occurs very frequently in our experiments because the backbone we use, OpenVLA-OFT, has weak capabilities and priors. As a result, it often fails to rollout correct trajectories, causing all sampled trajectories within a group to be incorrect and ultimately leading to the problem described above.
> >
> > Additionally I would like to point out the claim DAPO makes is that having batches with 0 gradient reduces the magnitude of the gradient update for the local batch itself and doesn't necessarily mean that this will lead to vanishing gradients in model weights when doing updates. The current statement in the paper is incorrect and I would appreciate if the authors would remove the claim or correct it with appropriate details.

---

> > > ### Author Response · Authors · 2025-11-28
> > >
> > > Dear Reviewer,
> > > Thank you very much for your recognition and your positive feedback! We are particularly grateful for your increased score, which greatly encourages us. We will further refine and polish the manuscript according to your suggestions to make it as strong as possible.

---

### Author Response · Authors · 2025-12-03
**Summary for Area Chair (Part 3)**

### Reviewer urcu (Score: 8 → 10, Confidence: 5)

**Reviewer Response:  I do think this work is excellent. I choose to update my score to 10. Hope your paper will be an oral paper.** (Before Bug)

**Concerns Raised:**

- W1-2: Confusing Figure 5.1 presentation.  Missing compute/throughput details.
- W3: RL fails when base success rate is nearly 0%.

**Our Responses:**

- W1-2: (W1) Revised figure to show consistent backgrounds for fair comparison. (W2) Added training time information: 12-24 hours for RoboTwin, 1-2 days for LIBERO on 8×A800 GPUs.
- W3: Acknowledged this is an inherent RL limitation with sparse rewards. Demonstrated that minimal data suffices for cold-start, and stronger models like π0.5 enable true plug-and-play RL with OOD experiments showing 7%→85.9% improvements.

---

## Additional Experiments Conducted During Rebuttal

**1. Extension to Diffusion-based VLAs:** We extended SimpleVLA-RL to π0.5, a diffusion-based VLA, by converting ODE to SDE for probability computation. Results show π0.5 improved from 71.6% to 91.3% (+19.7%) on RoboTwin 2.0. Notably, π0.5 used only 5% of OpenVLA-OFT's data yet achieved superior performance.

**2. Language Generalization:** We tested robustness using seen vs. unseen prompts with varied sentence structures. Results: Lift Pot (10%/8%→66%/62%), Stack Two Bowls (41%/38%→73%/66%). The minimal 4-7% gap confirms robust performance across diverse language variations.

**3. Out-of-Domain Task RL:** We applied RL on 3 OOD tasks absent from SFT training data. Results: Click Alarmclock (69.5%→81.3%), Stack Bowls Two (10.1%→48.4%), Grab Roller (7.0%→85.9%). This demonstrates substantial improvements under distributional shifts.

---

We sincerely appreciate the Area Chair’s substantial efforts during this challenging period. We have provided detailed responses to all reviewer comments to address every concern as thoroughly as possible. We hope that our summary will help clarify the overall discussion process and ease your workload.

Sincerely,

ICLR 2026 Conference Submission 16445 Authors

---

### Author Response · Authors · 2025-12-03
**Summary for Area Chair (Part 2)**

## Summary of Issues Addressed

### Reviewer qg2P (Score: 6 → 8,  Confidence: 4)

**Reviewer Response:** Acknowledged concerns were addressed and updated rating. (Before Bug)

**Concerns Raised:**

- W1-3 & Q1: Unclear experimental setup regarding initial checkpoints, discrepancy between reported OpenVLA-OFT results and original paper, compute/data budget comparison between SFT and RL, and citation needed for vanishing gradients claim.
- Q2: Limited methodological novelty

**Our Responses:**

- W1-3 & Q1:  (W1) We clarified that RL uses SFT-finetuned checkpoints. (W2) We explained that the result discrepancies arise from model simplifications (single-view images, LLaMA2 output head instead of MLP) detailed in Appendix F.2, and our results match the comparable configuration in the OFT paper Table 1. (W3) We provided a detailed compute comparison table showing SFT (~ 20 hours, 70k steps) vs RL (~30 hours total, ~60 steps) on 8×A800-80GB.  (Q1) We acknowledged the claim needs correction. The issue is about zero advantage when all samples fail, not inherent vanishing gradients.
- Q2: Please see the response to Reviewer qg2P, where a detailed clarification has been provided.

------

### Reviewer EvC5 (Score: 8, Confidence: 4, No response)

**Concerns Raised:**

- W1-Q1: (W1) Missing discussion of diffusion policy RL methods in related work. (Q1) Why SimpleVLA-RL works for diffusion-based VLAs?

**Our Responses:**

- We clarified that our OpenVLA-OFT is an autoregressive VLA.  However, our method is fully compatible with diffusion-based VLAs. In the rebuttal, we extend SimpleVLA-RL to π0.5 (a diffusion-based VLA using flow matching) by converting the ODE process to SDE for probability computation. Our experiments on RoboTwin 2.0 show π0.5 improved from 71.6% to 91.3% average success rate.
- Q1: We explained the gradient mechanism. For correct actions, maximizing log probability pulls predicted mean toward rewarded actions. For incorrect actions, minimizing log probability pushes the model away from penalized actions.

------

### Reviewer ogXC (Score: 6, Confidence: 4, No response)

**Concerns Raised:**

- W1: Limited methodological novelty
- W2: Justification for transferring LLM RL to VLA
- W3: KL regularization ablation and training stability
- W4: Real-world validation of "Pushcut" phenomenon
- Q1: Extension to continuous-action VLAs
- Q2: Sim-to-real gap examination
- Q3: Language generalization evaluation
- Q4: Training instability or performance oscillation
- Q5: RL training on out-of-domain tasks absent from SFT data

**Our Responses:**

- W1:  Please see the response to Reviewer ogXC, where a detailed clarification has been provided.
- W2:  We justified the transfer from two perspectives. In architecture, both LLMs and VLAs share Transformer-based backbones. In output paradigm, both involve sequential generation, LLMs output reasoning token sequences and VLAs output action sequences, with each token/action having an associated probability distribution.
- W3:  We added KL ablation studies in Appendix D.1  and showed curves about test success rate, gradient norms, loss, and sampling success rates. Results demonstrate that removing KL regularization leads to slight performance improvements while maintaining stable training.
- W4: We acknowledged that validating "Pushcut" in real-world requires online RL on physical robots, involving manual resets, human supervision, and higher costs. Due to these practical constraints, we placed this experiment in the Discussion section and plan comprehensive real-robot RL experiments in future work.
- Q1: We successfully extended SimpleVLA-RL to π0.5, a diffusion-based VLA using flow matching. By converting the ODE process to SDE, we enabled probability computation for RL. The results on RoboTwin 2.0 showed π0.5 improved from 71.6% to 91.3% average success rate.
- Q2: We provided real-robot test results showing the sim2real gap: “Stack Two Bowls” (76% sim →70% real), Click Bell (90% sim →60% real), Place Bottle (60% sim →14% real). While gaps exist, the key finding is that RL improvements achieved in simulation successfully transfer to real-world deployment.
- Q3: We conducted experiments with seen vs. unseen prompts. Results are Lift Pot (SFT: 10% seen /8% unseen, RL: 66% seen/62% unseen), Stack Two Bowls (SFT: 41%/38%, RL: 73%/66%). The RL-trained model maintains robust performance on unseen language instructions.
- Q4: We provided training curves in Appendix D.2 for both OpenVLA-OFT and π0.5 on RoboTwin 2.0, including loss, gradient norm, and sampling success rate. All curves exhibit smooth progression and convergence with no significant oscillations.
- Q5: We performed RL directly on 3 out-of-domain tasks not seen during SFT using π0.5. The results are Click Alarmclock (69.5%→81.3%), Stack Bowls Two (10.1%→48.4%), Grab Roller (7.0%→85.9%). This demonstrates SimpleVLA-RL achieves substantial improvements even under distributional shifts.

---

### Author Response · Authors · 2025-12-03
**Summary for Area Chair (Part 1)**

Dear Area Chair,

We sincerely appreciate your efforts during this challenging period. We understand the exceptional workload you're managing given the recent situations, and we're grateful for your careful consideration of our work.

To help you better understand the manuscript, we would like to provide a summary of our recent discussions with the reviewers.

---

## Discussion Overview

| Reviewer | Initial Rating | Revised Rating (Before Bug) | Confidence | Reply |
|---|---|---|---|---|
| qg2P     | 6              | 8             | 4          | yes |
| EvC5     | 8              | 8             | 4          | no |
| ogXC     | 6              | 6             | 4          | no |
| urcu      | 8               | 10           | 5          | yes |

Before the OpenReview bug occurred, **Reviewer qg2P** and **Reviewer urcu** had responded positively and raised their scores, bringing the average score to 8.0. Notably, although the **Reviewer EvC5** and **Reviewer ogXC** have not yet had the chance to further discuss or revise their scores, third-party statistics indicate that our updated average score of 8.0 already placed us in a tie for 10th among all ICLR submissions. Moreover, all reviewers reported a confidence level above 4, with the reviewer who gave a score of 10 reaching a confidence of 5. We are deeply grateful to all four reviewers for their high recognization of our work and their valuable feedback.

### Follow-up Review Details

1. **Reviewer qg2P** (Nov 26, 2025, 04:37; Score: 6 -> 8): Responded stating *"Thanks for the detailed response and additional results. **Overall I think this is a good paper with interesting experiments.** The response from authors have addressed my concerns so I updated my rating."*
2. **Reviewer urcu** (Nov 21, 2025, 17:50; Score: 8 -> 10): Responded stating *"Thank you for your efforts to address weaknesses. All of my concerns have been addressed…* ***I do think this work is excellent. I have updated my score to 10. I hope your paper will be selected as an oral presentation.**"*

As mentioned above, **Reviewer qg2P and Reviewer urcu** improve the score and clearly indicate that we have addressed all of their concerns. For the remaining two reviewers, the comments from **Reviewer ogXC** can be grouped into three groups. **W1, W2, and W4** focus primarily on the method, for which we have provided detailed clarifications. **W3 and Q4** are about some training dynamics and ablation studies, and we have now fully supplemented these experiments. **Q1, Q2, Q3, and Q5** are more open-ended questions, and for these we have provided a set of interesting empirical results that we believe sufficiently address the reviewer’s concerns. **Reviewer EvC5** is mainly interested in the implementation of VLA RL on diffusion models. In response, we have substantially expanded both the theoretical analysis and the experimental results in this direction. This additional content has also been positively acknowledged by **Reviewer qg2P**, and we believe it adequately addresses **Reviewer EvC5**’s concerns.

---

## Paper Strengths

**1. Simple yet effective method.** Our RL framework is remarkably simple and scalable. We use only sparse 0/1 outcome rewards based on task completion, completely eliminating the need for complex reward engineering. The training pipeline only requires loading the actor model (no critic, no reference model after removing KL regularization), and our exploration-enhancing strategies are straightforward to implement. Reviewer EvC5 noted this is *"extremely simple and scalable"* and *"the first work to demonstrate reliable improvement with RL for VLAs in simulated domains."* Reviewer urcu similarly praised the simplicity and reproducibility.

**2. Strong empirical results across multiple dimensions.** Despite the simplicity, SimpleVLA-RL achieves SOTA performance: 99% success rate on LIBERO and 80% relative improvement on RoboTwin 2.0. Beyond high performance, we demonstrate strong data efficiency, with only one demonstration per task, RL improves LIBERO-Long from 17.3% to 91.7%. Our generalization analysis shows RL improves unseen tasks while SFT suffers from catastrophic forgetting. Real-world experiments confirm successful sim-to-real transfer (17.5%→38.5%). Reviewers qg2P and EvC5 both acknowledged the significant performance improvements, strong generalization, and successful real-world transfer.

**3. Scalable and extensible VLA-RL framework.** We built SimpleVLA-RL on top of veRL with VLA-specific extensions: parallel multi-environment rendering for efficient sampling, while preserving veRL's distributed training optimizations. As demonstrated in our paper and rebuttal, the framework supports different simulation benchmarks (LIBERO, RoboTwin), different VLA architectures (autoregressive OpenVLA-OFT, diffusion-based π0.5), and different RL algorithms (GRPO, PPO).

---

### Meta-Review · Area_Chair_bGSu · 2026-01-07

**Summary:**

The paper introduces SimpleVLA-RL, a framework that employs reinforcement learning to Vision-Language-Action (VLA) models. The authors demonstrate that complex reward engineering and dense supervision are unnecessary for VLAs. Instead, a simple outcome-based approach can achieve State-of-the-Art performance on LIBERO and RoboTwin compared to supervised fine-tuning baselines. The reviewers initially questioned the novelty of applying known LLM techniques to robotics and whether the method was restricted to autoregressive architectures but exclude the increasingly popular diffusion-based policies. The primary concerns regarding novelty and architectural generality have been addressed during the rebuttal. All reviewers have reached a positive consensus.

**Reviewer Concerns:**

- Applicability to diffusion policies (EvC5, ogXC). The most significant concern was whether the method only worked for autoregressive models such as OpenVLA. In the rebuttal, the authors implemented and evaluated SimpleVLA-RL on a diffusion-based VLA ($\pi_{0.5}$). This experiment showed noticeable gains.

- Novelty (qg2P, ogXC). Reviewers questioned if the contribution was too simple. The author argued and demonstrated that simplicity is a feature for scalability and reproducibility, achieving results that complex prior methods could not.

- Baselines and details (qg2P). Discrepancies regarding OpenVLA-OFT baselines were resolved by clarifying architectural simplifications used for training efficiency.

- The cold start problem. RL requires a non-zero base success rate.

The first three concerns have been largely resolved in the rebuttal. However, the cold start problem remains a fundamental property of the method, though the authors showed that 1-shot SFT is sufficient to bootstrap learning.

**Reviewer Scores:**

Based on the discussions, urcu has agreed to increase the score from 8 to 10. qg2P will also likely to increase the rating. The other two reviewers have not replied. But since both have already provided a positive rating, they might or might not update their scores.

---

### Decision · Program_Chairs · 2026-01-26

Accept (Poster)